# Zinc-finger BED domains drive the formation of the active *Hermes* transpososome by asymmetric DNA binding

Laurie Lannes [1], Christopher M. Furman[1], Alison B. Hickman[1] & Fred Dyda [1] ✉

The *Hermes* DNA transposon is a member of the eukaryotic *hAT* superfamily, and its transposase forms a ring-shaped tetramer of dimers. Our investigation, combining biochemical, crystallography and cryo-electron microscopy, and in-cell assays, shows that the full-length Hermes octamer extensively interacts with its transposon left-end through multiple BED domains of three Hermes protomers contributed by three dimers explaining the role of the unusual higher-order assembly. By contrast, the right-end is bound to no BED domains at all. Thus, this work supports a model in which Hermes multimerizes to gather enough BED domains to find its left-end among the abundant genomic DNA, facilitating the subsequent interaction with the right-end.

Transposable elements (TE) are discrete genomic regions that can move from one position to another within genomes. TEs have been found in all eukaryotic kingdoms as both active and inactive forms and can make up a large portion of eukaryotic genomes[1]. TEs are a major force in the shaping of genomes and the evolution of species by establishing novel cellular functions and pathways[2]. Furthermore, the origin of some human diseases is linked to TEs[3,4].

Most of the eukaryotic class II or DNA transposons move by cut-and-paste transposition that consists of the excision of the transposon by generating double-stranded breaks (DSB) at the ends of the element and its reintegration elsewhere, typically without specificity (Fig. 1a insert). An autonomous transposon is delimited by two ends featuring terminal inverted repeats (TIRs), with one or more genes between them (Fig. 1b exemplified with *Hermes* transposon). One of these genes encodes the transposase, an enzyme that mobilizes the transposon by carrying out the necessary nuclease and trans-esterification activities (Fig. 1a exemplified with *Hermes*)[5]. The transposase typically assembles into a multimer that recognizes and brings together the TIRs by relying on site-specific DNA-binding domain(s) such as helix-turn-helix or zinc-finger domains, which can be either upstream or downstream of the catalytic domain[6]. After the formation of the transposition complex or transpososome, DSBs are created at the ends of the TIRs, liberating the transposon, and the assembly finds its target DNA to integrate the TE.

The *Hermes* DNA transposon, belonging to the largest cut-and-paste superfamily called *hAT*, was isolated from the genome of the housefly *Musca domestica*[7]. It has been characterized biochemically[8,9], and the three-dimensional structure of its N-truncated form with its catalytic core (hereafter referred to as "Δ-BED" Hermes) has been solved (Fig. 1c)[9–11]. The *Hermes* transposon is ~3 kb and is delimited by two dissimilar ends of ~450 bp referred to as the left-end (LE) and the right-end (RE)[7]. The ends are capped by two imperfect 17 bp TIRs that differ from each other by two swapped base pairs (Fig. 1b)[7]. The Hermes transposase shows a preference for 8 bp target sites with a 5'-nTnnnnAn-3' pattern such that staggered integration of the transposon across the target site generates 8 bp target site duplications (TSD) (Fig. 1a)[12].

The Hermes transposase is composed of 612 amino acids and is organized in four domains (Fig. 1c)[7,10,13]. In principle, two transposase protomers, each carrying one catalytic domain with its active site, assembled into a dimer should be sufficient to liberate two transposon ends and integrate them into a target site. In fact, a number of DNA transposases from both the prokaryotic and eukaryotic domains appear to work as dimers[14–18], suggesting that a dimer is a minimal oligomerization unit. However, there are also a few examples of higher-order oligomers. The bacteriophage MuA transposase forms a tetramer upon synapsing its ends[19,20], whereas the related retroviral integrases can form even higher-order oligomers, at least in vitro[21,22].

Architecturally, the Hermes transposase is unusual as it spontaneously forms a ring-shaped tetramer of dimers (Fig. 1c). The dimers tightly associate using an "intertwined" dimerization domain (DD; Fig. 1c). The quaternary structure is held together by the swapping over

[1]Laboratory of Molecular Biology, National Institute of Diabetes and Digestive and Kidney Diseases, National Institutes of Health, Bethesda, MD 20892, USA.
✉e-mail: fred.dyda@nih.gov

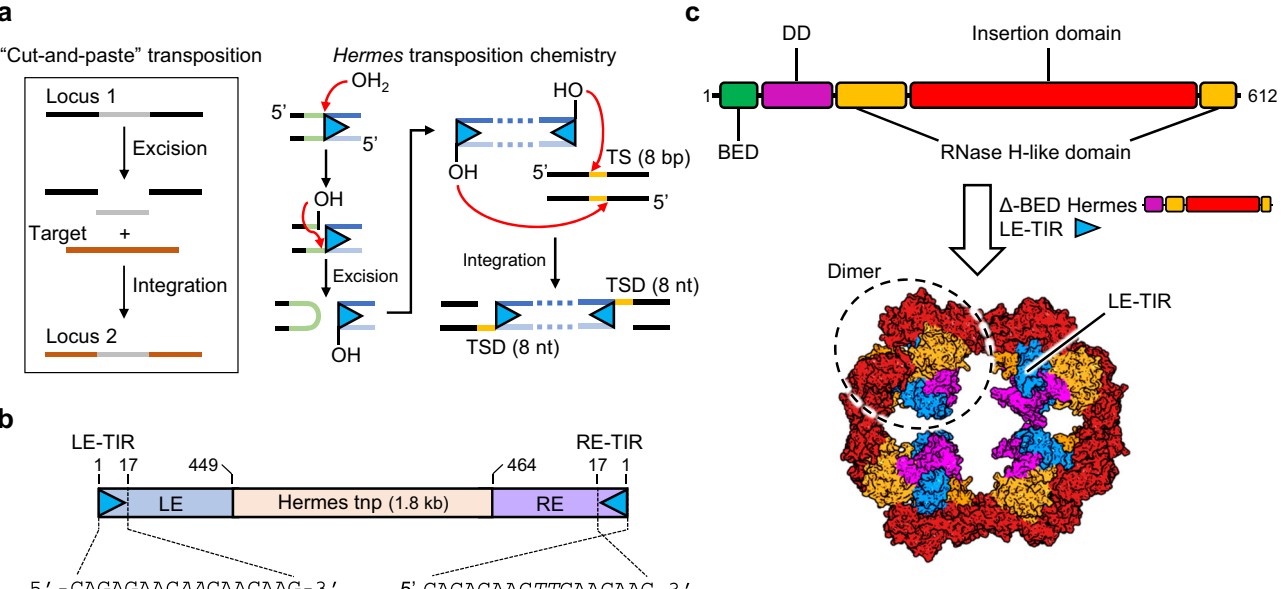

**Fig. 1 | The eukaryotic *Hermes* transposon system. a** The "cut-and-paste" transposition mechanism and *Hermes* transposon chemistry. The red arrows describe nucleophilic attacks on phosphate groups. TS target site. TSD target site duplication. **b** Schematic of the *Hermes* DNA transposon. The transposon has a left- and right-end (LE and RE, respectively) capped by two imperfect terminal inverted repeats (TIRs) that differ from one another by two swapped base pairs marked in italics. The ends flank the gene that encodes for the Hermes transposase (tnp). **c** Top: schematic of the domain organization of the Hermes transposase. DD dimerization domain. Bottom: octameric quaternary structure (tetramer of dimers) of BED-truncated (Δ-BED) Hermes in complex with the left-end terminal inverted repeat (LE-TIR) DNA (PDB: 4D1Q)[11].

of an α-helix from the insertion domain between adjacent dimers[10,11]. Deletion of the swapped helix prevents octamer formation, converting Hermes into dimers. Remarkably, these dimers are fully competent for all the enzymatic activities of Hermes in vitro; in fact, they are more active than wild-type Hermes. Yet they are inactive for transposition in cell-based assays[11]. The available biochemical and structural data provide no explanation for this substantial contradiction between in vitro and in-cell experiments.

All *hAT* transposases are predicted to contain an N-terminal BED zinc-finger domain organized around a conserved CCHH or CCHC motif that coordinates Zn$^{2+}$[13], but no structural information is available for Hermes as the constructs used in previous structural work lacked the first 78 amino acids. Nevertheless, this domain is indispensable for the excision step in vitro[11]. We have previously suggested that the BED domain of the Hermes transposase recognizes short subterminal repeats (STRs) interspersed in its transposon ends[11]. Even though the chemical steps of transposition occur at the very tips of TEs, several *hAT* transposons have DNA end requirements that span far beyond the TIRs. For instance, *Tol2* needs the terminal 200 and 150 bp of its LE and RE, respectively, for excision and transposition[23]; *Tag1* requires about 100 bp from both ends to generate comparable transposition rates as the full ends[24]; and *Ac* at least 200 bp from both ends[25]. In the case of *Tol2*, it has been shown that mutation of STRs reduced the excision activity[23]. Given the number of STRs in the *Hermes* ends, a dimer with only two BED domains does not appear sufficient if the binding of more than two is needed. Therefore, it is plausible to assume that interaction with multiple STRs is needed for in-cell activity, and that a higher-order assembly, such as the octamer with its eight BED domains, would be one way to supply a sufficient number of BED domains. However, the architecture of such an assembly is not known.

In this work, we aimed to decipher the role of the BED domain in transposon end binding and to understand how the transposase interacts with its ends. To explain the striking discrepancy between in vitro and in-cell results, we undertook a mechanistic study using biochemistry, X-ray crystallography, cryo-electron microscopy (cryo-EM), and in-cell transposition assays.

## Results

### The N-terminal domain of the Hermes transposase is a DNA-binding domain

Although the first 78 amino acids of Hermes that include the BED domain are indispensable for transposition in vivo and cleavage in vitro, they do not directly participate in recognition of the TIR or in the catalytic activity of the transposase[9,11]. Despite the critical role of the BED domain, its DNA-binding site has not yet been established. We, therefore, first sought to confirm the DNA-binding activity of Hermes' BED domain and determine if such activity was specific. We expressed the region 1-78 of the Hermes transposase (hereafter "BED") in *E. coli* and purified it (Supplementary Fig. 1a) to perform interaction assays. While the LE and RE of the *Hermes* transposon are 449 bp and 464 bp, respectively[7], the proximity in sequence space of the BED domain to the part of Hermes with known three-dimensional structure suggests that DNA binding would likely occur towards the interior portion of the TIR. This notion is also supported by previous activity data using mutated Hermes LE[11].

We used a double-stranded DNA (dsDNA) that spanned bp 11-27 of the LE (LE11-27, Fig. 2a) to probe the DNA binding of the BED domain. As a control, a randomized 17-mer (ran17) was used that had no common features with LE11-27. The DNAs were titrated with an increasing amount of protein from one to three equivalents and analyzed by size exclusion chromatography (SEC). The resulting chromatograms are presented in Fig. 2a. Binding was assessed by the shift of the DNA peak, the shape of any resulting complex peak, and whether we observed free DNA or protein as a function of titration. Although ran17 showed evidence for only weak binding by BED, LE11-27 formed a tight and stable complex that saturated at a 1:2 ratio of DNA to protein.

### The N-terminal domain of Hermes folds in a CCHC zinc-finger BED motif that specifically interacts with a subterminal repeat

Crystals were obtained of the 2:1 BED/LE11-27 complex that were ultimately optimized to diffract to 2.5 Å, and the structure was solved by zinc multiwavelength anomalous diffraction (Zn-MAD; Supplementary Table 1). The complex crystallized in space group P6$_5$22 with a

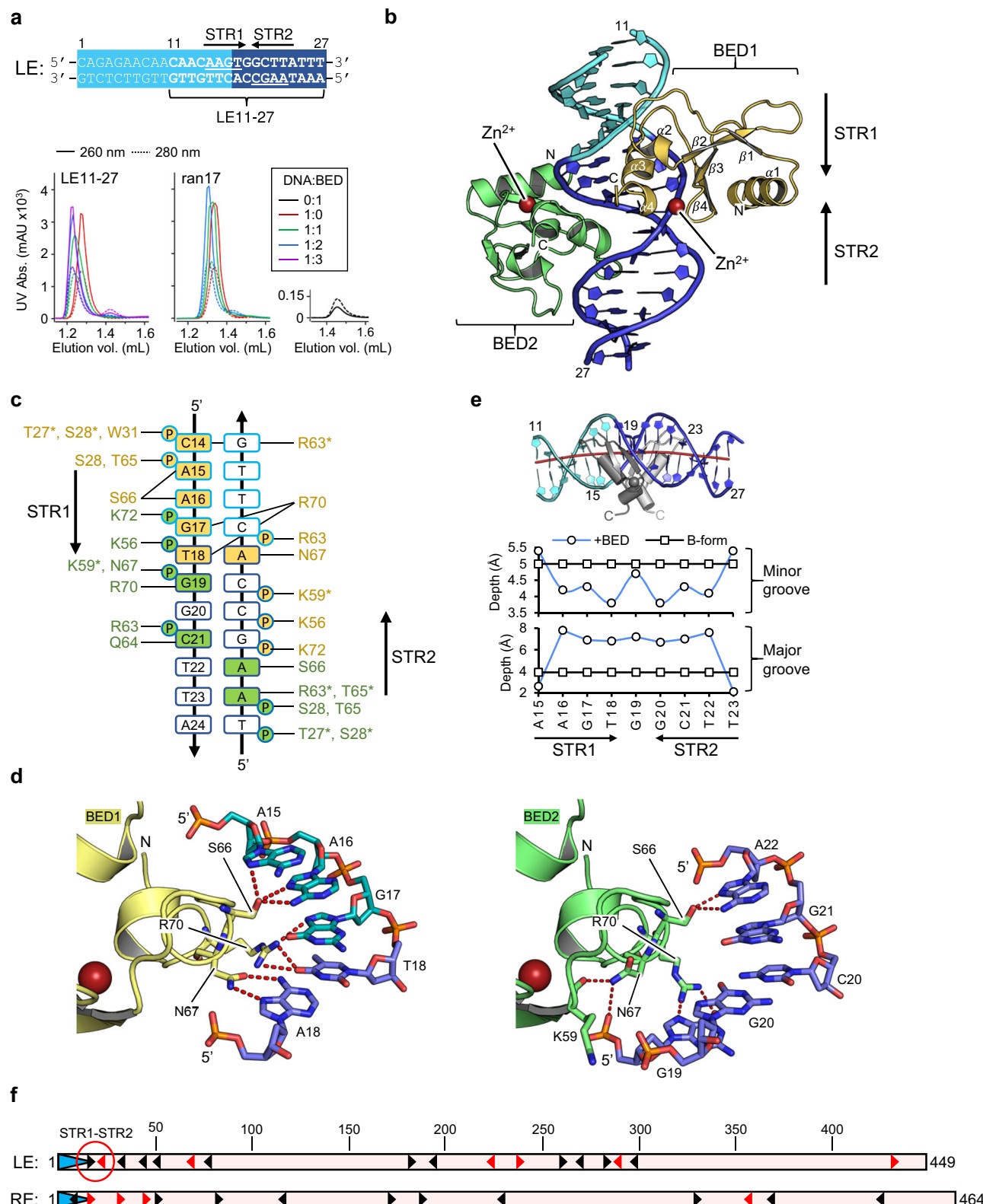

crystallographic twofold axis perpendicular to the oligonucleotide that intersected it at its central base pair. Thus, the asymmetric unit contained one BED domain and one DNA strand of the duplex. The blunt-ended double-stranded LE11-27 DNA used in crystallization was not perfectly palindromic and packed end-to-end in the crystal lattice. The crystallographic twofold axis, therefore, related the top half of the oligonucleotide to the bottom half, resulting in electron density that was the superposition of the two crystallographically related

orientations. Iterative model building and refinement led to the atomic model that is composed of two crystallographically identical BED domains bound to LE11-27. We modeled LE11-27 in the two overlapping and opposite orientations at 50% relative occupancy. We observed alternate conformations for some of the base-interacting amino acid side chains of the BED domain, consistent with the two orientations of LE11-27. This indicates that the BED domain has the ability to recognize the two halves of the imperfect palindrome by relying on these

**Fig. 2 | The N-terminal BED domain of the Hermes transposase. a** Sequence of the first 27 bp of the *Hermes* transposon left-end (LE). TIR terminal inverted repeat. The LE11-27 sequence was used as a binding substrate for the BED domain. The two subterminal repeats, STR1 and STR2 form a quasi-palindrome. The interaction was monitored by analytical size exclusion chromatography. **b** Crystal structure of two zinc-bound BED domains in complex with LE11-27 DNA. The DNA color code is the same as in **a** and the top strand is numbered. **c** Summary of the protein–DNA hydrogen bond network in the crystal structure. The amino acids in yellow and green belong to BED1 and BED2, respectively. The residues marked with a star interact via their main chain, while all the others engage their sidechain. The top strand of LE11-27 is numbered. The outline color of the bases follows the same code as in **a**, while the filling color identifies the BED domain interacting with them. Only

the phosphate groups (P) involved in hydrogen bonds with the proteins are pictured. The sugar puckers are not represented. **d** The hydrogen-bonding network (red dashed lines) between S66, N67, and R70 of the Hermes BED domain with the LE-STR1 (left) and STR2 (right) of the LE11-27 DNA. **e** Top: Curvature of LE11-27 DNA in the crystal structure. The color code for the DNA is the same as in **a**. The red line corresponds to the axis of the double helix obtained with Curves+[28]. For clarity, only residues 47–78 of the BED domain are shown. Bottom: Plots of the depth of the grooves of LE11-27 in B-form (black) and bound to BED (blue) at the top strand bases A15 to T23. **f** Mapping of the STRs 5'-AAGT-3' (black arrowheads) and 5'-AAGC-3' (red arrowheads) in the *Hermes* transposon LE and RE as putative BED binding sites. The TIRs are pictured as blue arrows at the start of each end.

---

alternate side-chain conformations. The atomic model for the whole assembly is shown in Fig. 2b.

Each BED domain contains at its C-terminal a CCHC zinc-finger involving residues C51, C54, H71, and C73 that coordinate a bound $Zn^{2+}$ ion. The two BED domains bound to the imperfect palindrome do not contact each other, but each C-terminal α-helix between T65 and R70 (α3; see Fig. 2b) is deeply inserted into the major groove. According to a DALI search[26], the only identifiable structural homolog that binds $Zn^{2+}$ is the BED domain of the human transcription factor ZBED2 protein, whose structure was determined by NMR (ZBED2; PDB 2DJR, RIKEN Structural Genomics/Proteomics Initiative, 2006). This protein was recently implicated in the development of lineage plasticity in pancreatic cancer cells[27]. The summary of the protein–DNA interactions is presented in Fig. 2c.

Key to binding the imperfect palindrome is the ability of the N67/R70 pair of the BED domain to assume alternate conformations. These, together with S66, Q64, and the main chain carbonyl of R63 and T65 at the apical loop (between β2 and α3) of the zinc-finger, form all the base-specific interactions. In one half of the palindrome, N67 of BED1 (shown in yellow, Fig. 2d and Supplementary Fig. 2a with the electron density) interacts with the Hoogsteen face of A18, while R70 interacts with the stacked G17 and T18. In the other half (BED2, shown in green, Fig. 2d and Supplementary Fig. 2a), the position that was occupied by A18 is now replaced by G20, which is less competent to interact with N67. We observe that N67 is completely displaced, no longer facing the DNA but instead interacting with K59 and the phosphate of G19, with R70 now in its place to interact with G19. For both BED domains, other contributions to LE11-27 binding include several nonspecific contacts between protein side chains of Lys, Arg, Ser, Thr, and Trp residues and the DNA phosphate backbone. Thus, the structure shows that the single BED domain of the Hermes monomer is able to form specific interactions with two related but distinct DNA motifs.

Analysis of the DNA parameters by Curves+[28] indicated several distortions relative to B-form DNA. The LE11-27 DNA experiences a slight bending of the double helix that is reflected in the positive roll and negative slide of base pair dyads (Supplementary Fig. 2b). The bending is directed towards the edges of the major groove and the C-termini of the BED domains (Fig. 2e). The minor groove shows widening over the base pairs flanking the central base pair (i.e., A16, G17, T18 and G20, C21, T22, respectively) of ~2 Å compared to B-form, whereas its major groove is narrowed around the central base pair before widening ~2 Å at the termini (Supplementary Fig. 2b). On the other hand, the major groove is deeper by ~4 Å over the region A16-T22, while the minor groove is shallower over this region (Fig. 2e). This suggests that BED binding to the imperfect palindrome may be allosteric, in which the insertion of one α-helix into the major groove and the resulting DNA distortion makes the binding of the second BED easier.

### The mapping of the STRs
The interactions observed in the crystal structure of the BED/LE11-27 complex suggests that the BED domain binding site is 5'-AAG(T/C)-3'.

There are 17 and 14 occurrences of this motif in the LE and RE of *Hermes*, respectively, concentrated in particular towards the tips of the transposon, with five copies in the first 55 base pairs of both ends (Fig. 2f and Supplementary Fig. 3a). We designate these putative BED binding sites as subterminal repeats (STRs), and number them from "STR1" onward moving in from the LE transposon tip but from STR0 on the RE as the first AAGT motif is within the TIR. With the exception of LE-STR1/RE-STR1 and the pair LE-STR3/RE-STR2, the rest of the STRs of the two ends do not align with each other: their arrangement is asymmetric at the two ends. Interestingly, on the LE, except for STR1, STR2 to STR7 are all oriented antisense, while RE-STR1 to RE-STR5 are all oriented in the sense direction. Beyond LE-STR1/LE-STR2, only one other pair of STRs is organized as a palindrome, LE-STR14/LE-STR15, far interior in the LE beginning at bp 276.

### Reconstitution of the Hermes transpososome in vitro
As our attempts to crystallize either the full-length Hermes transposase or its complexes with transposon ends had failed, we turned to cryo-EM to determine the architecture of the transpososome, the assembly containing two bound transposon ends. Although the LE and RE TIRs only differ from each other by two base pairs which do not interact with the protein, yet the transposase has a much higher affinity for the LE compared to the RE[8,11]. Therefore, to reconstitute the transpososome in vitro, we evaluated the interaction of various LE-DNA oligonucleotides complexed with the purified transposase to identify the best substrate for structural studies. We tested a range of oligonucleotides that extended the LE-TIR into the transposon, such as LE-TIR + 13 and LE-TIR + 30 as well as those that were nicked and gapped at the non-transferred strand at the cleavage site but also extended into the flanking DNA, i.e., 8 + LE-TIR + 7 and 8 + LE-TIR + 30 (Supplementary Fig. 4a and Fig. 3b). Samples were prepared by mixing purified Hermes transposase with sub-stoichiometric or stoichiometric amounts of DNA and dialyzed to lower salt concentration prior to analysis by SEC and mass photometry (MP).

The SEC elution profiles corresponding to the titration of Hermes with the various LE-DNAs are presented in Supplementary Fig. 4b–g. The results were consistent with the crystal structure of the Δ-BED-transpososome[11], in that close to one LE-TIR bound per protomer (Supplementary Fig. 4e). However, for longer DNAs, the stoichiometry changed, with a maximum of 0.5 DNA bound per Hermes protomer (Supplementary Fig. 4c, d, f, g). LE-TIR + 13, LE-TIR + 30, and 8 + LE-TIR + 30 led to precipitation of the complex when added to the protein in a stoichiometric 1:1 mix (Supplementary Fig. 4c, d, g), in contrast to 8 + LE-TIR + 7 whose complex remained stable but did not bind more DNA in going from a 1:0.5 to 1:1 mix (Supplementary Fig. 4f).

The peaks corresponding to the protein–DNA complexes were not monodisperse. This was also evident during the purification of the transposase, where we similarly observed heterogeneity and unusual behavior when the samples were analyzed by SEC. As shown in Supplementary Fig. 4h, the purified transposase itself eluted as two populations, P1 and P2, where the relative ratio between the two

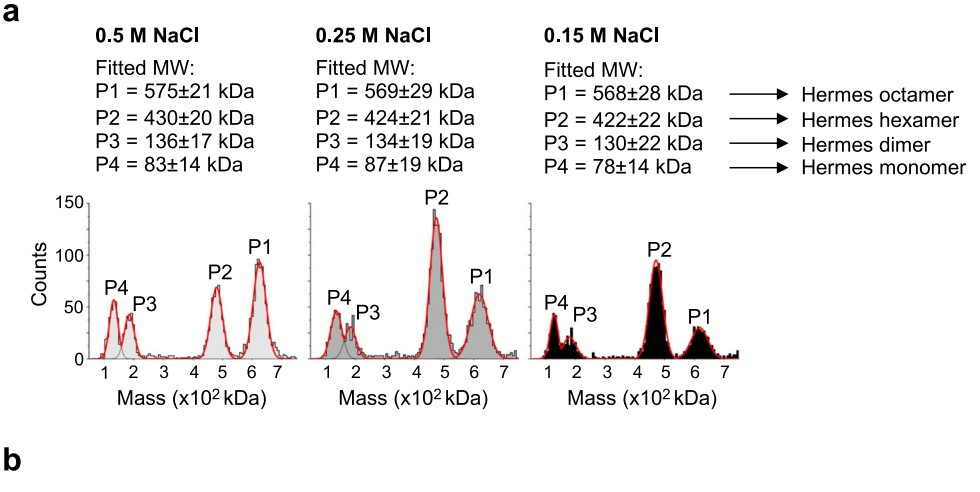

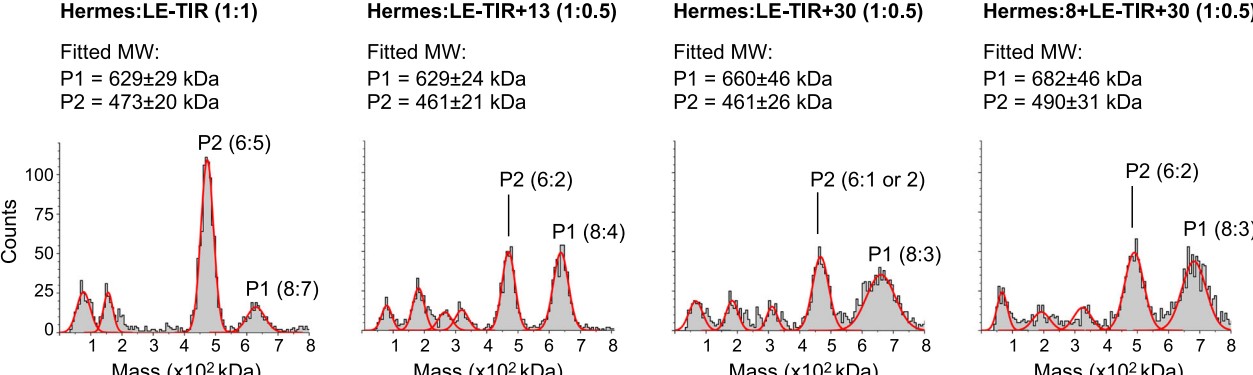

**Fig. 3 | Stoichiometry of Hermes/LE-DNA transpososomes. a** Mass photometry data of the purified Hermes transposase dialyzed against three different buffers containing 0.5, 0.25, and 0.15 M NaCl. The theoretical mass of a Hermes monomer, dimer, hexamer, and octamer are 70, 140, 421, and 561 kDa, respectively. **b** Mass photometry data of the Hermes/LE-DNA complexes at 0.15 M NaCl. The sequences of the top strand of the double-stranded DNAs are presented on top. The bases in gray were absent on the top strand, but their complementary bases were present on the bottom strand. The subterminal repeats STR1 to STR4 are underlined and their orientation is indicated by arrows. The Hermes-to-DNA ratio is given on top of each plot. The fitted masses of P1 and P2 are reported on top of each plot, and their molecular composition (Hermes:DNA) are reported inside each plot. The theoretical MW of a Hermes hexamer and octamer are 421 and 561 kDa, respectively.

elution peaks varied as a function of the salt concentration in the buffer, with the higher concentration favoring P1 (a species consistent with an octamer) and lower concentrations promoting P2 (a species consistent with a hexamer). Through a combination of SEC (Supplementary Fig. 4h) and MP analysis (Fig. 3a), we established that the Hermes transposase exists in solution as two oligomeric states composed of six or eight monomers.

We used MP (Fig. 3b) to measure the composition of the DNA-bound complexes observed by SEC. These results were consistent with the SEC data showing that the Hermes transposase assemblies bound fewer molecules as the DNA was lengthened. For example, the MP experiments revealed that LE-TIR was mainly bound to the hexameric form of Hermes and that approximately five DNAs were bound. The longer oligonucleotides also generated more heterogeneous samples with the appearance of several masses under 400 kDa and broad distributions that led to poor fitting (standard error >30 kDa). Nevertheless, under our experimental condition, the Hermes hexamer could bind up to two of either LE-TIR + 13 and 8 + LE-TIR + 30 DNAs; the

Hermes octamer bound up to four LE-TIR + 13, and three LE-TIR + 30 and 8 + LE-TIR + 30 DNAs.

**Three LE-STRs bind to BED domains within the transpososome**
On the basis of the SEC and MP experiments, for cryo-EM studies we selected the 8 + LE-TIR + 30 oligonucleotide as it contained both flanking region and the longest transposon DNA. It formed both hexameric and octameric complexes. As we assumed that in a biologically relevant assembly, two transposon ends are bound, we minimized the loading of the transposase and used a ratio of 1:0.25 protein-to-DNA. We also anticipated that the TIRs would be synapsed inside the same Hermes dimer given that the transposase integrates its transposon across an 8-bp target site to generate 8 bp TSDs, consistent with the distance between the two active sites in one Hermes dimer. The resulting complex was purified by SEC and the fraction with the highest absorbance was used for structure determination.

As shown in Fig. 4a, the 2D classes of the top view of the complex resembled an octamer, as observed in the Δ-BED-transpososome

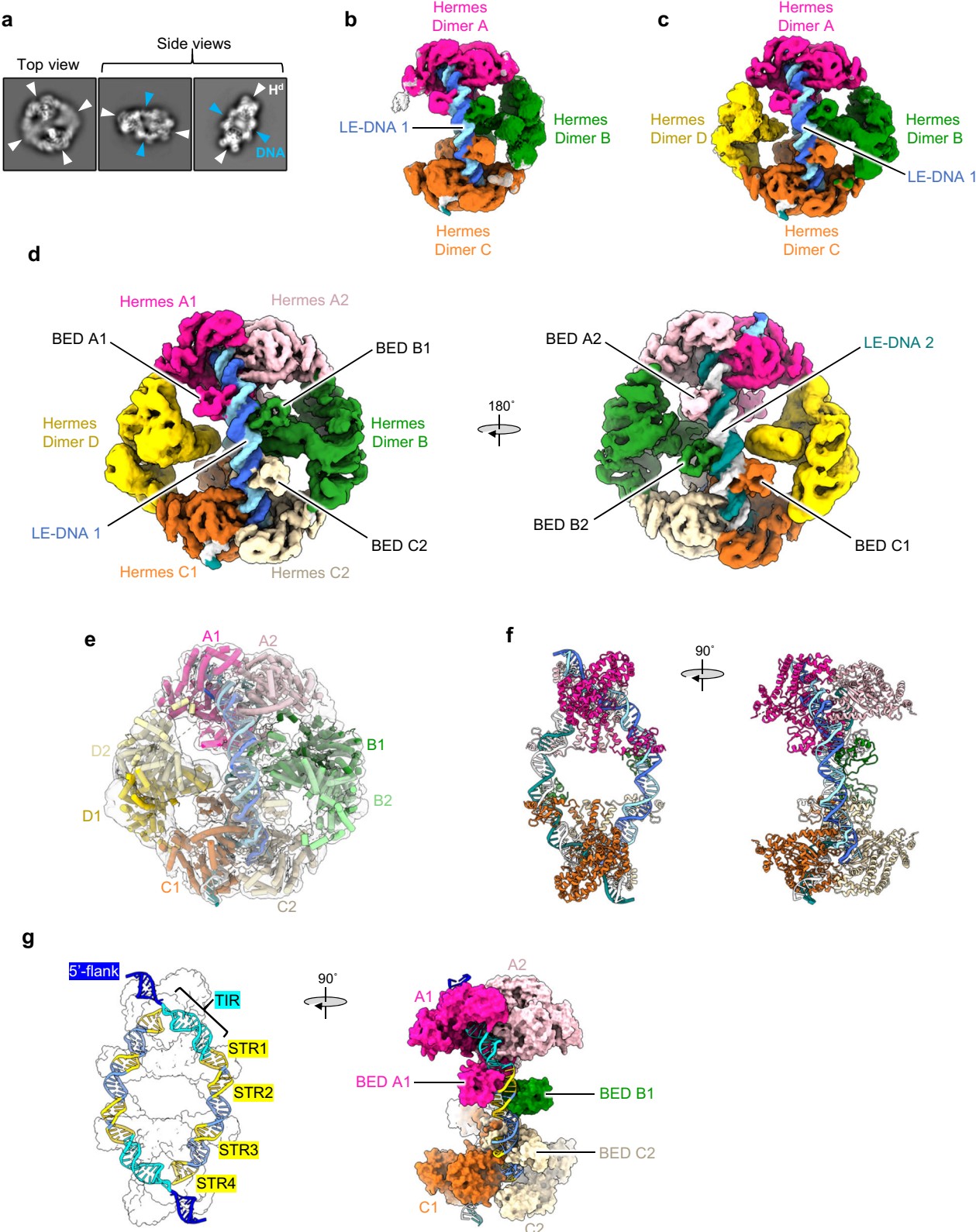

crystal structure[11], with resolved features for two transposase dimers facing each other and linked by two equatorial dimers to form a closed ring. While the top views did not give clear indication for the presence of DNA, in the side views, bound DNA was clearly visible. The particles were further cleaned by 3D classification, and the refined maps of their best classes are shown in Fig. 4b, c, respectively (see details in Supplementary Fig. 5). The reconstruction (from two 3D classes

representing 37% of the total particles) was composed of a closed octameric ring (Fig. 4c). On the other hand, the shape of the reconstruction presented in Fig. 4b from one 3D class that represented 19% of the particles was consistent with an octameric transpososome that appeared to have lost one of its equatorial dimers as no potential density was observed for it. Importantly, both maps indicated that two DNAs bridged two Hermes dimers, the top dimer A and the bottom

**Fig. 4 | The cryo-EM structure of the LE-LE Hermes transpososome. a** Example of selected 2D classes of the transpososome from the data processing in RELION[74–76]. The transposase dimers (H[d]) are marked with white arrowheads and the DNAs are indicated with blue arrowheads. The box size is 280 Å. **b** Cryo-EM map of the transpososome composed of three Hermes dimers (A, B, and C) and of two LE-DNAs (6.3 Å resolution). **c** Cryo-EM map of the transpososome composed of four Hermes dimers (A, B, C, and D) and of two LE-DNAs (5.5 Å resolution). **d** Multi-body refinement composite map of the transpososome. Body 1 corresponds to the DNAs, the Hermes dimers A and C and the BED domains of dimer B (4.6 Å resolution). Body 2 and Body 3 correspond to Hermes dimers B and D, and were refined in two

other maps at 10.2 and 10.9 Å resolution, respectively. **e** Model of the LE-LE Hermes transpososome inside the composite cryo-EM map. **f, g** The atomic model of the core of the LE-LE Hermes transpososome (PDB: 8EDG, this publication). **g** Left: the antiparallel LE-DNAs inside the complex (only the silhouette of the atomic surface of the Hermes dimers is shown). The features of the DNAs (TIR and STRs) are highlighted. Right: the atomic surface of the Hermes dimers is displayed while the LE-DNA 1 is shown in cartoon mode to emphasize the extended protein/DNA interaction inside the transpososome. All the maps presented were sharpened and denoised with DeepEMhancer[85]. The corresponding RELION's post-processed maps are presented in Supplementary Figs. 5–8.

dimer C. The maps aligned well and displayed the same features for these two dimers. In contrast, the equatorial transposase dimers (dimers B and D) were poorly resolved and featureless in both reconstructions, likely due to the sparsity of interactions with the DNAs (see local resolution mapping in Supplementary Fig. 5).

In order to resolve the interactions between the BED domains and DNA, we performed multi-body 3D refinement on the three best 3D classes (regardless of the oligomerization state of the transposase) using masks to divide the transpososome into three bodies. Body 1 was composed of the DNAs, the Hermes dimers A and C and the BED domains originating from the equatorial Hermes dimers bound to the DNAs, and the other two corresponded to the N-truncated Hermes dimers B and D, respectively. The resulting composite map is presented in Fig. 4d. The map for Body 1 was refined at 4.64 Å resolution, while the other two maps were of much lower resolution (Body 2, 10.21 Å; Body 3, 10.94 Å, Supplementary Fig. 6). The resolution of the Body 1 map was sufficient to perform model building (Fig. 4f, Supplementary Fig. 8, maps-model FSC curves are reported in Supplementary Fig. 7). In conjunction with rigid-body fitting of several copies of the crystal structure of the apo Δ-BED Hermes monomer (PDB: 2BW3)[10], we obtained the structure presented in Fig. 4e.

The analysis of the final composite map in Fig. 4d revealed that two Hermes dimers A and C (with each monomer A1/A2 and C1/C2 in pink and orange, respectively) are bridged by two DNA duplexes. There are also two DNA-bound BED domains (in green) that have been contributed by the B dimer. Thus, three BED domains bind to each DNA. It was possible to unambiguously assign the orientation of the DNA molecules (hereafter 5′ and 3′ always refer to the top strand) since the 5′-flanking region, as well as the two-nucleotide gap on the top strand of the 8 + LE-TIR + 30 DNA, were clearly visible. The two LE-DNAs are antiparallel, with each TIR interacting with a different Hermes dimer next to the 3′-end of the other oligonucleotide (Fig. 4f, g). The configuration of the transpososome with two antiparallel DNAs was the only configuration we saw, yet such an arrangement with the TIRs bound by opposite dimers cannot support the integration of the Hermes transposon across an 8 bp target site.

The 8 + TIR-LE + 30 DNA includes the TIR and STR1 through STR4. STR1 is clearly bound by the BED domain that belongs to the same Hermes protomer (A1 or C1) that also binds to the TIR as indicated by distinct linker density between the BED domain and the dimerization domain (DD in Fig. 1c). The equatorial BED domains bound to STR2 are closest to dimer B whereas dimer D is further away on the other side of the DNAs. Furthermore, the STR2 motifs are also bound in the hexameric Hermes transpososome where only one equatorial dimer is present at the position of dimer B. Therefore, we conclude that the equatorial Hermes dimer B engages both its BED domains with the STR2 motifs. The final map also showed weak density that linked the STR3-bound BED domains to the core of protomers A2 or C2. STR4 was not bound by any BED domain; rather, the 3′-end of each oligonucleotide (bp 36–47) was positioned similarly as the TIR entering the catalytic domain of the A2 or C2 protomer, with bp 47 sitting at the place of the TIR bp 3. The insertion of the 3′-ends inside a Hermes

dimer in a TIR-like manner was unexpected as it appears to place a full-length transposon end at risk if it were to be inadvertently cleaved. However, the DNA sequence at the oligonucleotide 3′-end has no similarity to that of the TIRs. Of note, no protein-protein interactions between the BED domains or between the BED domains and the rest of their transposase were observed.

## The STRs of the RE do not interact with the BED domains inside the transpososome

As we were unable to generate interpretable cryo-EM data for a single Hermes assembly that bound one LE and RE transposon end at the same time, we determined the structure of the transpososome bound to two REs. Using a similar approach as described for the LE-LE version based on partial signal subtraction to focus the reconstruction on the DNAs and the Hermes dimers A and C (Supplementary Fig. 9), a reconstruction at 5.1 Å resolution was obtained and a model could be generated (Fig. 5e and Supplementary Fig. 10). The 3D class that refined best clearly showed 5′-flanks synapsed in the top A1/A2 dimer revealing two parallel DNAs. In marked contrast to the LE-LE transpososome, none of the refined 3D classes showed density corresponding to either free or bound BED domains (Supplementary Fig. 9). The 3′-ends of 8 + RE-TIR + 30 DNAs were both inserted in the C1/C2 dimer in a TIR-like fashion, suggesting that this interaction was not a function of STR/BED interactions, but a possible consequence of the particular length of the DNA used.

## The minimal BED binding site is the AAGT motif, but it must be followed by an AT-rich region

It was very surprising that in the RE-RE complex, no BED domain was bound to RE-STR1 as we had anticipated it would be bound as seen for LE-STR1. This forced us to reevaluate the notion that the AAG(T/C) motif indeed represents the BED domain recognition site. We first asked whether RE11-27, that contains RE-STR1 (5′-AAGC), binds the BED domain (Supplementary Fig. 11). SEC analysis showed that RE11-27 DNA binds similarly as the randomized control, ran17, and did not form a stable complex. Likewise, a mutant RE11-27 (RE11-27T) with an AAGT motif replacing AAGC did not form a stable complex either. However, on the LE, LE11-27mut, where STR2 (5′-AAGC) was replaced by 5′-AAAA, formed a stable 1:1 protein-to-DNA complex. These results suggested that an isolated AAGT motif was necessary but not sufficient for BED binding.

We noted that the AAGT motif in LE11-27mut was flanked by two AT-rich regions, whereas RE11-27T presented the AT-rich region only at its 5′-end. We then tested if a BED domain interacts with an isolated AAGT motif only if it is followed by an AT-rich region (Supplementary Fig. 11, LE11-27mut-5′G vs. LE11-27mut-5′3′G), and we found this was indeed the case. We verified whether it was also the case for an isolated AAGC motif, as the RE-STR1 did not show binding. The SEC data obtained with LE11-27mutC showed only weak binding comparable to ran17, suggesting that an isolated AAGC motif was not sufficient. Collectively, these experiments revealed that the AAGT motif is the effective BED binding site, but it must be accompanied by an AT-rich region to achieve high-affinity DNA binding.

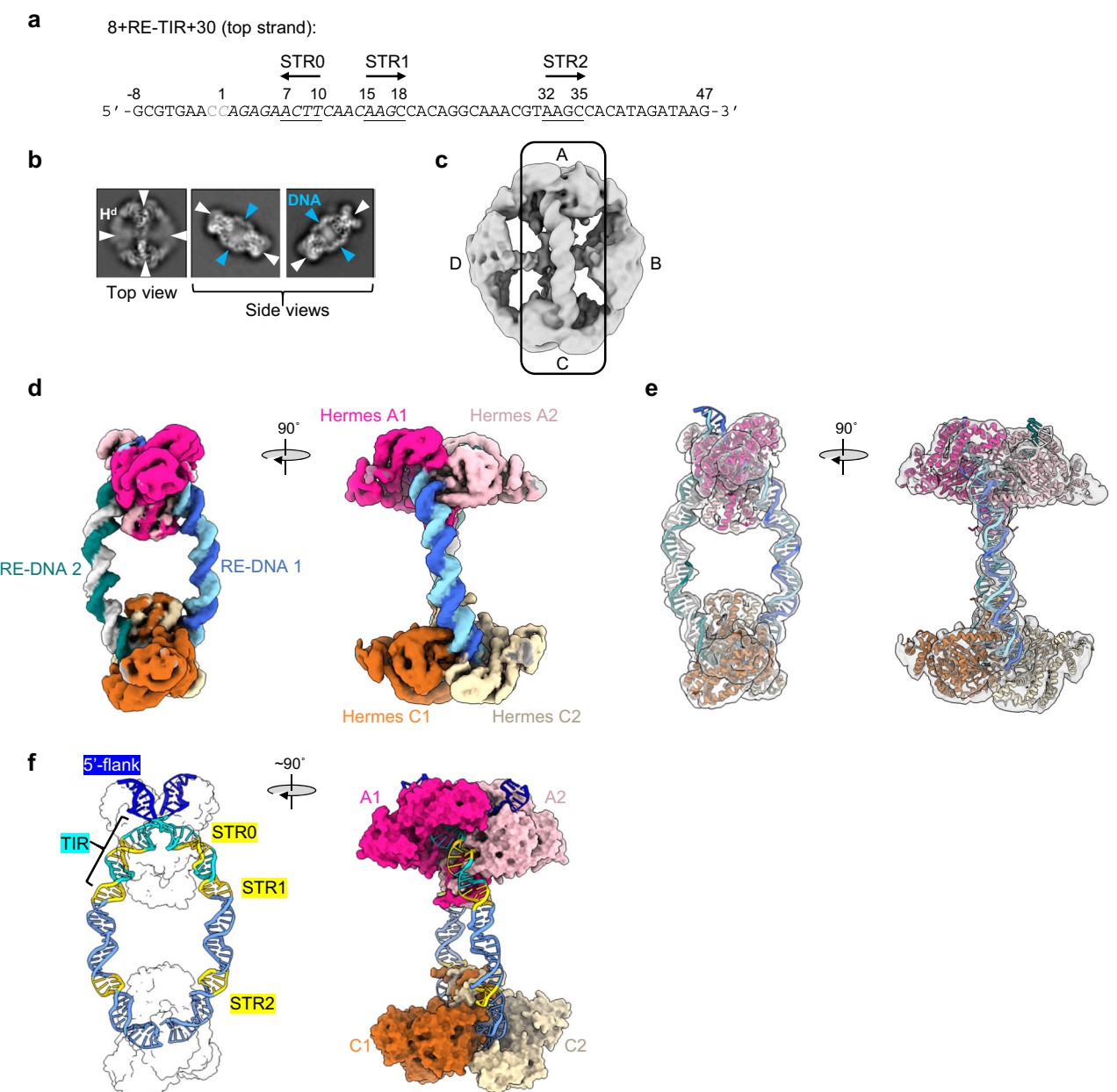

**Fig. 5 | The cryo-EM structure of the RE-RE *Hermes* transpososome. a** Sequence of the top strand of the 8 + TIR-RE + 30 DNA (RE-DNA). The bases in gray were absent on the top strand, but their complements were present in the bottom strand. The terminal inverted repeat (TIR) is in italic, the subterminal repeats STR0 to STR2 are underlined, and their orientation is indicated by arrows. **b** Example of selected 2D classes (box size is 250 Å) from the data processing in RELION[74–76]. The transposase dimers (H$^d$) are marked with white arrowheads and the RE-DNAs are indicated with blue arrowheads. **c** The best 3D class of the transpososome is composed of four Hermes dimers marked A to D, and two RE-DNAs. **d** The map of the core of the RE-RE *Hermes* transpososome (5.1 Å resolution). No clear density for DNA-bound or free BED domains is present. The map was sharpened and denoised with DeepEMhancer[85]. The corresponding RELION's post-processed map is presented in Supplementary Figs. 9, 10. **e, f** Atomic model of the core of the RE-RE *Hermes* transpososome inside its cryo-EM map. **f** Left: the parallel RE-DNAs inside the complex (only the silhouette of the atomic surface of the Hermes dimers is shown). The features of the DNAs (TIR and STRs) are highlighted. Right: the atomic surface of the Hermes dimers is displayed, while the LE-DNA 1 is shown in cartoon mode.

## The interaction of the BED domains with the quasi-palindromic LE-STR1-STR2 is cooperative

The characterization of the BED binding site suggested that the AAGC motif might need the presence of the AAGT motif adjacent to it in a palindromic configuration for optimal binding. Furthermore, the crystal structure suggested the possibility of cooperative binding between the two BED domains as the two α3 helices from two protomers are inserted into a deepened major groove adjacent to each other. To test this hypothesis, we used electrophoretic mobility shift assays (EMSA) with fluorescently labeled LE11-27 and LE11-27mut DNAs

and the BED domain. The resulting polyacrylamide gels are shown in Fig. 6b. As expected, the DNAs generated two different delayed bands resulting from the interaction of two and one BED with LE11-27 and LE11-27mut, respectively. No band corresponding to a single binding event, even in the early stage of the titration, was detected for LE11-27. Therefore, only simultaneous double interaction events were observed consistent with cooperative binding. As the crystal structure of the BED/DNA complex showed no evidence of protein-protein contacts, it appears therefore, that cooperativity is the result of DNA deformation.

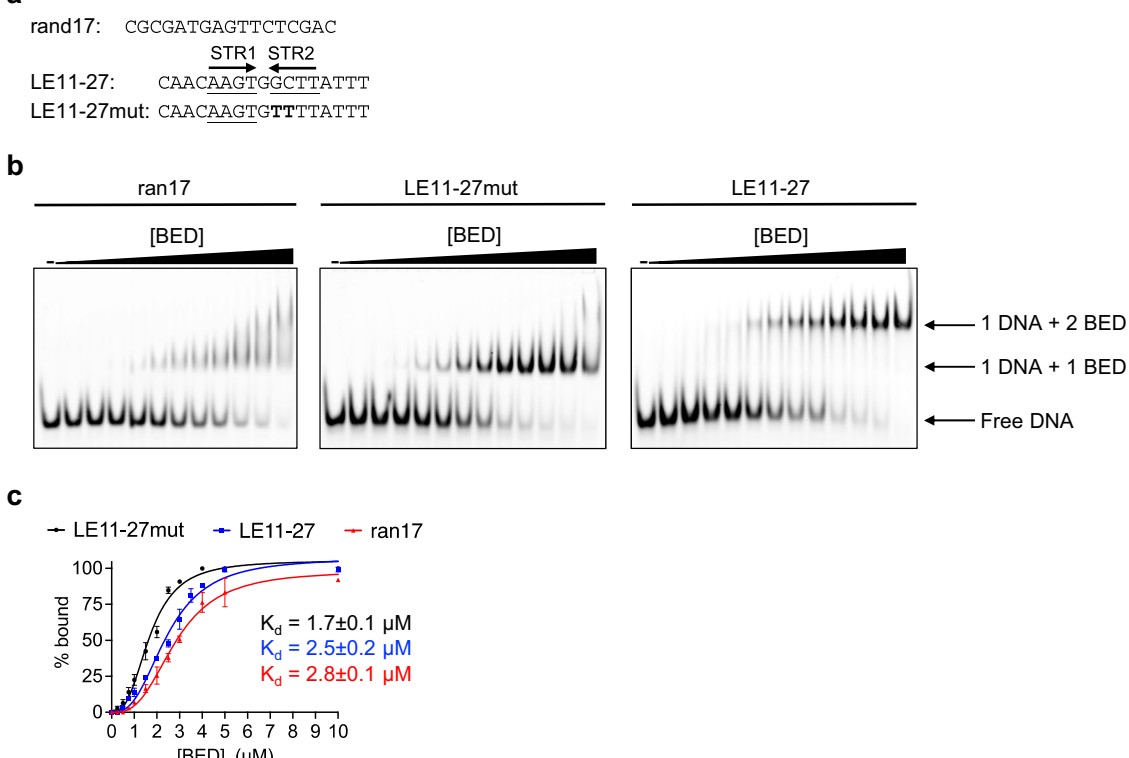

**Fig. 6 | Cooperative interaction of the BED domain with *Hermes* left-end (LE) STR1-STR2 motif. a** The sequence of the top strand of the DNAs ran17, LE11-27mut, and LE11-27 double-stranded DNAs. The subterminal repeats STR1 and STR2 are underlined, and their orientation is indicated with arrows. The mutated bases in LE11-27mut are in bold. Ran17 is the unspecific interaction control. **b** The electrophoretic mobility shift assay (EMSA) gels (15% polyacrylamide, 0.5X TBE). The DNAs (1 μM) were titrated with increasing amounts of BED protein from 0 (-) to 10 μM. **c** Binding curves of ran17 (red), LE11-27mut (black), and LE11-27 (blue) with the BED domain derived from EMSAs performed in triplicate (Supplementary Fig. 12). Values are mean ± standard deviations. The dissociation constants ($K_d$) from the fitting are reported for each DNA sequence.

## Restoring asymmetric BED binding sites in the LE/LE transposon does not rescue transposition activity in cells

The cryo-EM reconstructions indicated that the LE/LE pair, in contrast with the RE/RE pair, is in antiparallel orientation, a configuration that cannot support the integration of the transposon. We suspected that the BED/STR interactions within the LE/LE pair were responsible for the antiparallel orientation. Previously, we showed that a modified *Hermes* transposon with symmetrized LE/LE ends were inactive in insect cells[11]. We asked if we could rescue the transposition of a LE/LE transposon carrying a puromycin marker, by mutating the first three STRs of one end to make it resemble the RE, contained on a LEmut donor plasmid, p-donor (Fig. 7b). The transposase was expressed from a helper plasmid, p-helper. Both plasmids were transfected into HEK293T cells. After several days under puromycin selection, colonies were counted as a read-out (Fig. 7a). As *Hermes* transposition in HEK293T cells has not been previously reported, we first established it was active on its wild-type ends. As a comparison, *Hermes* showed a higher transposition rate than that of wild-type *piggyBac* under identical transfection conditions. (*Hermes* was also reported as highly active in *S. pombe*, so the T317A mutation had to be introduced to tame it)[12]. Symmetrized LE/LE p-donor led to a drastic loss of mobilization in *Drosophila* S2 DEV8 cells[11], and similarly, in HEK293T cells, LE/LE p-donor retained little mobilization, while the RE/RE transposon showed no activity. Finally, the LE/LEmut p-donor showed the same activity as the LE/LE p-donor. Thus, restoring the STR asymmetry in the first 50 bp in one of the LEs was not sufficient to rescue the mobilization of the LE/LE transposon. Like previous experiments in insect cells, transposition activity in HEK293T cells also required substantially longer transposon ends. Complexes assembled with such ends were insoluble, so we have

thus far been unable to study them with structural tools. However, it appears clear that the asymmetry of the transposon ends is an important requirement for activity.

## The minimal end requirement for *Hermes* transposition in HEK293T cells

The transposition assay showed that the divergence of sequence between the ends must span beyond the first 50 bp to have a mobile system. Hence, we sought to establish how much of the LE is needed to observe transposition while keeping the RE unchanged. We tested nine different LE/RE p-donors wherein the LE was gradually truncated (LE50 to LE306) as well as one in which the region spanning over the first three STRs was deleted (LEΔ), and the results are presented in Fig. 7c. The respective lack of significant integration of the transposons LE50/RE and LEΔ/RE confirmed that neither the 50 bp downstream of the LE-TIR nor the rest of the LE are sufficient for mobilization in cells when combined with the full RE.

The truncated LE constructs showed that mobility equivalent to the full-length LE was retained up to 223 bp. A slight decline in integration started at 190 bp to finally reached a drastic downturn at 140 bp and shorter. The loss of transposition activity was steady, allowing us to affirm that the critical truncation point on the LE lies between 140 and 190 bp when associated with the full-length RE. Experiments with partially scrambled LE sequences suggest that there is no influence on transposition by not keeping constant the distance between the LE-TIR and the CMV enhancer (Fig. 7e). Similarly, we also asked what the minimal end requirement was for the RE when combined with LE223 (Fig. 7d). We observed that the RE truncated after 327 bp resulted in similar integration as the full-length RE. The

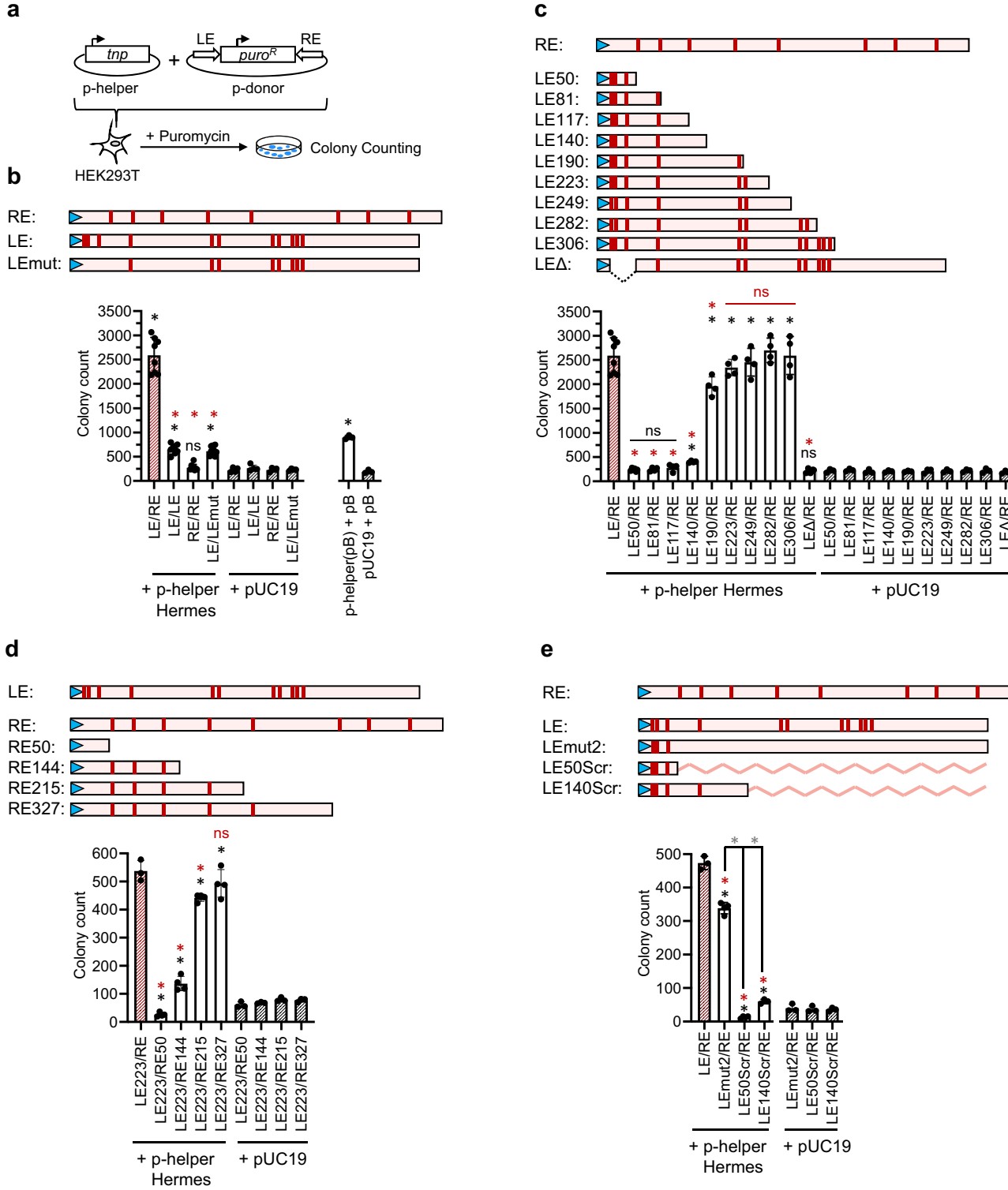

transposition activity was slightly reduced with 215 bp of RE, while it is severely impaired when only 144 bp of RE were retained. We conclude that the *Hermes* transposon needs at least ~220 terminal bp of both its LE and RE to be effectively mobilized in cells.

**The binding of auxiliary Hermes transposases inside the transposon ends to support transposition in cells is unlikely**

In the subterminal regions of the LE and RE that are critical for activity, there are several STRs that are putative BED binding sites. On the LE, there are two putative BED binding sites between bp 140 and 223 at

positions 183 and 190 (LE-STR5 and LE-STR6). There are also two STRs on the RE between bp 144 and 327 at positions 172 and 228 (RE-STR4 and RE-STR5). To determine if these STRs are important for *Hermes* transposition in cells, we tested the mobility of a transposon donor in which the LE putative BED binding sites were all mutated (LEmut2/RE, Fig. 7e). We also tested donors in which the LE sequence was scrambled after bp 50 (LE50Scr/RE) or after bp 140 (LE140Scr/RE). Whereas LEmut2/RE retained ~70% of wild-type activity, the scrambled LE sequences in LE50Scr/RE and LE140Scr/RE p-donors showed no or little mobility (Fig. 7e). It, therefore, appears that the putative LE BED

**Fig. 7 | *Hermes* transposition assay in HEK293T cells. a** Schematic of the experimental procedure. The plasmid p-helper encoding the Hermes transposase (tnp) and the p-donor containing the puromycin resistance gene *puro^R* (1.5 kb) flanked by the *Hermes* transposon left-end (LE) and right-end (RE) are transfected into HEK293T cells. The cells that experience a chromosomal integration of the *LE-puro^R-RE* transposon are selected against puromycin and the colonies are counted. **b–e** Top: Schematic of the *Hermes* LE, RE, and their variants (truncation or mutations) used in the assay. The terminal inverted repeats (TIRs) are represented by blue arrowheads and the subterminal repeats putative Hermes BED domain binding sites are depicted as red bars. Bottom: Histograms of the colony count for each p-donor. The combination of the transposon ends is reported on the x-axis for each p-donor. Each black circle represents one independent data point. The mean values are plotted as columns (red-stripped and black-stripped columns for the controls and white columns for the experimental data) the standard deviations are represented with bars. In **b** $n = 8$ (experimental data and LE/RE) or $n = 4$ (pUC19 controls) or $n = 3$ (pB). In **c–e** $n = 4$ (experimental data) or $n = 3$ (controls). Positive controls (red-stripped columns) $n = 8$ in **b**, **c**, $n = 3$ in **d**, **e**. pUC19 controls $n = 4$ (in **b**) or 3 (in **c–e**). The statistical two-tailed unpaired t-test was applied to determine whether the experimental data were statistically different from the LE/RE (in **b**, **c**, **e**) or LE223/RE (in **d**) experiment (annotated with a red star when significant or a red "ns" when not) and their pUC19 controls (annotated with a black star when significant or a black "ns" when not), respectively. In **b** *$p < 0.0001$ and ns$p = 0.2478$. In **c** red*$p < 0.0001$ except for LE190/LE red*$p = 0.0096$, red ns$p = 0.2357$; 0.5290; 0.6032, and 0.9953 (from left to right, respectively), black*$p < 0.0001$ except for LE306/LE black*$p = 0.0002$, black ns$p = 0.3997$; 0.6441; 0.1525; 0.4557 (from left to right, respectively). In **d** red*$p < 0.0001$ except for RE215/LE223 red*$p = 0.0047$, red ns$p = 0.2549$, black*$p < 0.0001$, black ns$p = 0.0036$ and 0.0066 (from left to right, respectively). In **e** *$p < 0.0001$ except for LEmut2 red*$p = 0.0002$, LEScr50 black*$p = 0.0036$ and LEScr140 black*$p = 0.0082$. The *piggyBac* transposition system (pB) was used as a general control of the experimental design.

binding motifs downstream of the first 50 bp are not needed for transposition, ruling out a model in which auxiliary transposases besides the octameric transpososome are required. Nonetheless, it appears that some as-yet-unidentified feature inside the *Hermes* transposon ends is needed to support transposition in cells.

## Discussion

The cut-and-paste *Hermes* transposon stands out among the transposases that have been biochemically or structurally characterized so far. While most DNA transposases function as dimers or tetramers, Hermes is unique that it forms closed octameric rings as a tetramer of dimers. While a mutant version of Hermes, incapable of forming octameric rings, that forms only dimers, is hyperactive for transposition in vitro at low salt conditions, it is inactive at physiological ionic strengths or in cells[11]. We sought to establish a mechanistic explanation of the higher-order organization necessary for in-cell activity.

The cryo-EM structures of the cores of the LE-LE and RE-RE transpososomes determined here provide a new framework for understanding the role of Hermes' unique ring-shaped organization. In the LE-LE complex, although the two ends were antiparallel (an arrangement that clearly cannot support transposition), each of the first three STRs within the first 35 bp of the LE interacted with a BED domain originating from three different Hermes dimers (Fig. 8a, left). This included the high-affinity LE-STR1-STR2 site that is present only on the LE of *Hermes*. The RE adopted a similar but parallel positioning within the transpososome, but it did not interact with BED domains (Fig. 8a, right). Taken collectively, we propose that transposition activity in cells relies on a transpososome assembly that can supply a sufficient number of BED domains and, in the correct configuration, first recognize and tightly bind to its transposon LE. Seemingly, a ring-shaped octamer that assembles a stable closed-form oligomer has evolved to serve this purpose. Furthermore, the binding of an LE, as it interacts with three dimers out of four, could shape the three-dimensional organization of the Hermes assembly such that it might not be able to accommodate three STR/BED interactions with a second LE bound parallel to the first. Thus, the antiparallel LE binding we observed might be the most energetically stable solution for the Hermes assembly in the presence of multiple LEs. We propose that the cooperative binding of two BED domains to the LE-STR1-STR2 palindrome is the key to the recognition of the LE and this is the first binding event (Fig. 8b). The cryo-EM structure of the core of the RE-RE transpososome showed that the RE does not interact with any BED domain, suggesting that RE binding relies on the interaction of its TIR with one dimer in the octamer, perhaps aided by nonspecific contributions from dimer C across the octamer. Assisting the process is the fact that once the LE is bound, the RE is only a few kb away at the other end of the transposon (Fig. 8b).

The LE-STR/BED interaction relies on a rigid pattern with the STR motifs precisely positioned relative to one another and to the TIR,

compatible with the three-dimensional arrangement of the BED domains inside the transposase octamer. For example, the cooperative binding of two BED domains with the LE-STR1 and LE-STR2 is only possible due to their palindromic arrangement, and we have previously shown that even a single bp shift in the position of the LE-STR1 and LE-STR2 away from the LE-TIR severely impairs the transposon cleavage activity[11]. These observations suggest that the exact phasing of the LE-STRs is crucial. Certainly, the closed architecture of a ring-shaped assembly is more rigid when compared to the alternative of linear arrangement, and we suggest that this is an important aspect of supporting the proper spatial organization of the BED domains and their interactions with the LE-STRs. It is possible that the role of the equatorial dimers is to provide BED domains to the complex and hold the ring together so that additional nonspecific interactions can form with the dimer that is diametrically opposite to the catalytically active dimer.

Conceptually, the accumulation of zinc-finger (ZF) motifs for high-affinity binding is similar to that of transcription factors (TFs) that are composed of ZF arrays. For example, the essential insulator protein CTCF has eleven ZFs[29], and KRAB-ZF proteins feature twelve ZFs on average[30]. The ZFs of these polydactyl proteins recognize a wide variety of DNA motifs of typically three or four bases avoiding redundancy within the same protein. The organization in arrays enables the combination of ZFs to interact with longer target sequences and increases the affinity of the protein toward DNA sites. As the Hermes transposase did not evolve the luxury of multiple BED domains arranged as beads on a string as did ZF-TFs, it apparently solved its affinity problem by high-order multimerization to make an array out of a single domain on the polypeptide. The use of a short recognition motif has a problem, though, as it is present throughout the genome. Thus, while the 5′-AAGT motif is the essence of the minimal BED binding site, high-affinity interaction occurs only if an AT-rich region follows. AT-rich sequences tend to be intrinsically bent, and we observed that in our crystal structure of the BED/DNA complex, three lysines interact with the phosphate group of three nucleobases positioned at the 3′-end of the specific binding sites. This interaction seems to be facilitated by the bending of the DNA, and in the case of an AAGT motif followed by an AT-rich sequence, it is plausible that the intrinsic bending of the 3′-end of the motif favors the interaction of these three lysines, thereby stabilizing BED binding.

It was previously suggested that the *Hermes* STR was 5′-GTGGC as it was repeated several times close to the TIRs, and also found in the closely related *hobo* transposon[11,31]. Our binding data suggests that this should be reevaluated, as 5′-AAGT represents the BED binding site, although its binding mode is complex as it also depends on the sequence following the tetranucleotide. The Hermes BED domain has some nonspecific DNA-binding affinity but reaches high affinity specific binding only if the AAGT motif is followed by an AT-rich sequence. Under these constraints, *Hermes* has at most eight other putative BED

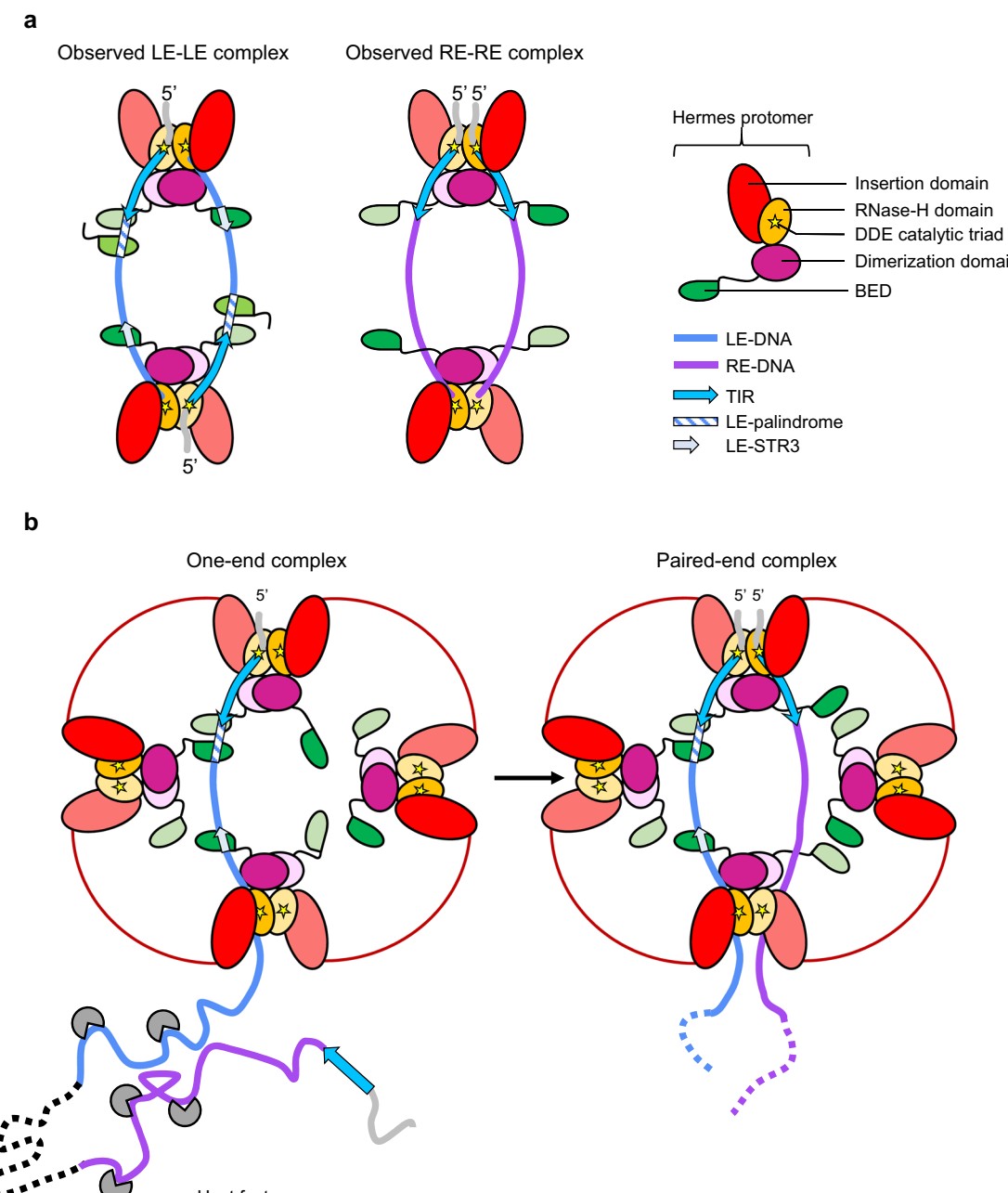

**Fig. 8 | Model of the formation of the *Hermes* transpososome in cells.**
**a** Schematic of the organization of the core of the two-left-end (LE-LE) and two-right-end (RE-RE) *Hermes* transpososomes observed by cryo-EM. In the LE-LE transpososome, the LE-TIR interacts with the catalytic center of one Hermes protomer, the LE-STR1-STR2 palindrome and the LE-STR3 interact with three BED domains from three Hermes protomers belonging to three dimers, and the LE-DNA 3'-end interacts with the opposite Hermes dimer. The LE-DNAs interact with the transposase assembly in an antiparallel orientation. In the RE-RE transpososome, the RE-TIR interacts with the catalytic center of one Hermes protomer, and the RE-DNA 3'-end interacts with the opposite Hermes dimer. The RE-DNA does not interact with BED domains. The RE-DNAs interact with the transposase assembly in a parallel orientation, with both RE-TIRs present in the same Hermes dimer.
**b** Proposed model of the formation of the *Hermes* transpososome in cells. The LE is first recognized by the transposase assembly, because of its higher affinity conferred by its quasi-palindromic STR1-STR2 motif (one-end complex). Once the LE is bound, the RE is only a few kilobase away at the other end of the transposon and can then be recognized solely based on the sequence of its TIR. The unspecific interaction of the 3'-end of both LE and RE might support the formation of the transpososome. Host factor DNA-bending proteins may interact with the transposon to facilitate end pairing. When both TIRs are synapsed inside the same Hermes dimer (paired-end complex), the transpososome is in the proper configuration to perform its cut-and-paste transposition chemistry.

binding sites on the LE after bp 50, and eight on the RE (Supplementary Fig. 3b). The *Hermes* ~200 bp minimal ends requirement for in-cell transposition strongly suggests that the sequence beyond the terminal 50 bp of both ends is involved in the formation of a competent transpososome yet mutation of the putative BED binding sites inside the LE did not dramatically reduce transposition activity. This suggests that the binding of additional Hermes transposases is not required in cells, and instead, we hypothesize that a cellular factor may be involved in directing the formation of the synaptic complex in HEK293T cells. For example, DNA-bending proteins have been identified as transposition cofactors both in prokaryotes and eukaryotes. The bacterial HU protein was reported to associate with bacteriophage *Mu*[20,32], and the

IHF protein with *Tn10*[33], *Tn4652*[34], and the Tn7-like transposon from the type I-F *Vibrio cholerae* CRISPR-associated transposase system[35]. In mammalian cells, the HMGB1 protein interacts with *Sleeping Beauty*[36], as well as the recombination signal sequence (23RSS) of the RAG1/RAG2 V(D)J recombination complex closely related to transposition[37,38]. By bending the DNA, these proteins are proposed to facilitate the formation of the synaptic complex. We have not been able to identify a candidate motif inside the *Hermes* transposon ends: HMGB1, for example, does not have a well-defined binding motif; rather, it interacts with non-canonical and pre-bent DNA[39,40]. However, the *Hermes* LE and RE have an AT-content of 64 and 70%, respectively, with a high potential for intrinsic bending. A better understanding of the influence of host cell factors on the mobility of the *Hermes* transposon would be of great interest, especially in the scope of the development of genome editing tools based on *Hermes* and other *hAT* superfamily transposons.

Our results have implications beyond *Hermes* to other members of the *hAT* superfamily. Kunze and coworkers have reported that the N-terminal region of the *hAT* Activator (Ac) transposase recognizes several STRs scattered in both ends of *Ac* in a cooperative manner[41–43]. We suggest that it is the cooperative binding of Ac's $C_2H_2$-BED domains that might be responsible for this. Most of the *hAT* transposons have sets of STRs close to the TIRs[44]. Interestingly, the *hAT Tol2* transposon presents five 5′-AAGT motifs in the first 50 bp of its LE and RE, and the LE somewhat resembles *Hermes* LE, featuring a perfect palindrome (5′-AAGTACTT) directly after the TIR followed by a third antisense STR 10 bp downstream[45]. Furthermore, all the *hAT* transposases have a BED domain at their N-terminus[44]. However, to our knowledge Hermes is the only *hAT* transposase that was expressed and purified to obtain soluble transposase; therefore, biochemical and biophysical data involving the crucial question of multimerization by other *hAT* transposases are unfortunately not currently available. It has been suggested that Tol2, Tgf2, and the domesticated Kat1 can form oligomers (bigger than dimers) spontaneously or upon DNA binding[46–48]. Therefore, it is possible that the ability to deploy a multitude of BED domains and to cooperatively bind to STRs might be generalizable to other *hAT* transposons as well.

The *hAT* transposons have been co-opted several times as a source of the coding sequence for the emergence of new genes and functions[44]. Even though domesticated proteins have evolved from intact transposases, it seems that their enzymatic activity is rarely retained, contrary to their DNA-binding capacity[49,50]. The *hAT* BED domain has been conserved in many cases and even sometimes replicated, as in the vertebrate ZBED4 and ZBED6 proteins that have four and two BED domains, respectively[51]. The ZBED transcription factors are expressed in various vertebrate tissues and have been found to be involved in the regulation of many functions, such as the expression of ribosomal protein genes[52], embryogenesis and carcinogenesis[51], retinal morphogenesis[53], or muscle development[54]. Our structural results might help in better understanding how these proteins function.

The simplicity of cut-and-paste transposon systems makes them appealing to re-purpose as genomic tools as transposons are naturally occurring genome editing systems. Importantly, cut-and-paste DNA transposition does not require extensive DNA repair and, therefore can work in cell lines where repair mechanisms are not efficient. Currently, the two most widely used transposon systems to modify mammalian genomes are *piggyBac* and *Sleeping Beauty*[55,56]. However, as wild-type transposases typically have suboptimal activity since they are not under positive selection, this gives us the opportunity to generate hyperactive transposons. A spectacular example of this possibility is *Sleeping Beauty*, whose transposase was first reactivated and then at least two orders of magnitude of increased transposition activity was achieved using educated guesses from sequence alignment and randomized genetic screens[57,58]. The *Hermes* transposon is mobile in a variety of organisms from yeasts[59–61], to various insects[62–65], and here we showed also in human tissue culture and at activity levels comparable to *piggyBac*. The understanding of the principles of transposome assembly and organization at the three-dimensional level should open up possibilities for the rational redesign of Hermes to join the array of transposon-based tools available for the modification of mammalian genomes.

## Methods

### Isolated BED domain expression and purification
The region coding for the N-terminal BED domain of Hermes (Uniprot Q25442) spanning from residues 1 to 78 (BED) was cloned into a pET15b expression vector in such a way that the recombinant protein was not tagged. The protein BED was expressed in Rosetta2(DE3) *E. coli* cells (Novagen) by growth at 37 °C in LB medium supplemented with 50 µg/mL carbenicillin until reaching $OD_{600nm}$ ~ 0.6, followed by cooling at 16 °C and induction by addition of IPTG to the final concentration of 0.25 mM. After 16 h, 1 L of culture was harvested by centrifugation and resuspended in 50 mM Tris-HCl pH 7.5, 100 mM NaCl, 0.3 mM TCEP, 2.5 mM $MgCl_2$, DNaseI, and protease inhibitors (Roche). The cells were disrupted by sonication and the soluble part of the lysate was purified on a 5 mL-HiTrap heparin column (GE Healthcare) at 4 °C. The column was equilibrated with 25 mM HEPES.Na pH 7.5, 100 mM NaCl, 0.3 mM TCEP, and protease inhibitors. BED was eluted with a linear salt gradient (100 mM to 1 M NaCl). The fractions that contained BED were concentrated and further purified at 4 °C on a preparative Superdex 75 16/600 size exclusion column (GE Healthcare) equilibrated with 25 mM HEPES.Na pH 7.5, 500 mM NaCl, and 0.3 mM TCEP (see chromatogram in Supplementary Fig. 1a). The protein and its purity were checked at each step of the purification by SDS-PAGE electrophoresis (4–12% Bis-Tris NuPAGE, MES as running buffer, Invitrogen) stained with SimplyBlue SafeStain (Novex) (Supplementary Fig. 1a).

### Hermes transposase expression and purification
The region coding for the full-length Hermes transposase (Uniprot Q25442), with the mutations Q2E and K128G was cloned into pBAD/Myc-His (Invitrogen) in such a way that the recombinant protein was not tagged. Hermes was expressed in Top10 (Invitrogen) *E. coli* cells by growth at 37 ˚C in LB medium supplemented with 50 µg/mL carbenicillin until reaching $OD_{600nm}$ ~ 0.6, then followed by cooling at 16 ˚C and induction by addition of arabinose to the final concentration of 0.01%. After 16 h, 4 L of culture were harvested by centrifugation and resuspended in 50 mM Tris-HCl pH 7.5, 500 mM NaCl, 0.3 mM TCEP, 40 µM $MgCl_2$, DNaseI, and protease inhibitors (Roche). The cells were disrupted by sonication and the soluble part of the lysate was purified on two 5 mL-HiTrap heparin columns (GE Healthcare) at 4 ˚C. The column was equilibrated with 25 mM HEPES.Na pH 7.5, 100 mM NaCl, 0.3 mM TCEP, and protease inhibitors. Hermes was eluted with a linear salt gradient (640 mM to 1 M NaCl). The fractions that contained the protein were concentrated and further purified at 4 ˚C on a preparative Superose 6 XK 16/70 size exclusion column (GE Healthcare) equilibrated with 25 mM HEPES.Na pH 7.5, 750 mM NaCl, 0.3 mM TCEP, and protease inhibitors (see chromatogram in Supplementary Fig. 1b). The fractions containing Hermes were then concentrated to be directly used or frozen in liquid nitrogen to be stored at −80 ˚C after the addition of glycerol to a final concentration of 15%. Hermes eluted as a peak with a low molecular weight shoulder; we excluded the fractions corresponding to the shoulder (Supplementary Fig. 1b).

### Preparation of the double-stranded DNA samples
All the oligonucleotides were purchased from Integrated DNA Technologies (IDT). The lyophilized DNAs were dissolved in 10 mM Tris-HCl

pH 8.0 to a concentration of 1 mM or 1.5 mM for the unlabeled DNAs and to a concentration of 100 μM for the 6FAM-labeled DNAs. The double-stranded DNA samples were prepared by mixing stoichiometrically the complementary strands to a final concentration of 500 μM (unlabeled) or 10 μM (6FAM-labeled). The mixes were heated to 95 °C for 10 min and slowly cooled to room temperature, and stored at −20 °C. All the DNAs used in this study are reported in Supplementary Table 2.

### BED/DNA interaction assay monitored by size exclusion chromatography (SEC)

Purified BED was dialyzed overnight at 4 °C against 25 mM HEPES.Na pH 7.5, 150 mM NaCl, 0.3 mM TCEP, and protease inhibitors (Roche) and subsequently concentrated (Vivaspin 20 3 kDa MWCO, GE Healthcare) to ~7 mg/mL. The samples (60 μL) were prepared by mixing the protein and the oligonucleotide from the 500 μM stock solution in various ratios (1:0, 0:1, 1:1, 2:1, and 3:1 DNA-to-protein ratio with 1 equivalent corresponding to 100 μM) and equilibrated on ice for at least 15 min. About 10 μL of the sample (10 μl injection loop loaded with 50 μL sample) were injected on a Superdex 75 PC 3.2/30 (GE Healthcare) analytical size exclusion column preequilibrated with the protein buffer. The UV absorbance at 280 and 260 nm were monitored to identify the different species.

### Hermes transposase stoichiometry determined by SEC

Purified Hermes at 1 mg/mL in B750 buffer (750 mM NaCl, 25 mM HEPES.Na pH 7.5, 0.3 mM TCEP, and protease inhibitors) was either dialyzed overnight or 3 h against buffer B500 (500 mM NaCl, 25 mM HEPES.Na pH 7.5 and 0.3 mM TCEP). The latter sample was then transferred in B250 (250 mM NaCl, 25 mM HEPES.Na pH 7.5, and 0.3 mM TCEP) either overnight or for 3 h. The latter sample was finally transferred to B150 (150 mM NaCl, 25 mM HEPES.Na pH 7.5, and 0.3 mM TCEP) for 16 h. The three samples were analyzed by SEC (Superose 6 30/100, GE Healthcare) in their respective buffer. The absorbance at 280 nm was monitored. Calibration curves (MW as a function of the elution volume) in B500, B250, and B150 were obtained, using three proteins from the Sigma-Aldrich Gel Filtration Markers kit: thyroglobulin (669 kDa), apoferritin (443 kDa), and β-amylase (200 kDa).

### Hermes/DNA interaction assay monitored by SEC

Purified Hermes was concentrated (Vivaspin 20 50 kDa MWCO, GE Healthcare) to ~3 mg/mL. The samples were prepared by mixing the protein and the double-stranded DNAs from the 500 μM stock solution in various ratios (1:0, 0:0.25, 1:0.25, and 1:0.5 protein-to-DNA ratio with 1 equivalent corresponding to 34 μM). The samples were successively dialyzed at 4 °C against 500, 250, and 150 mM NaCl containing buffers (25 mM HEPES.Na pH 7.5, 0.3 mM TCEP, and protease inhibitors) for 2, 4 h, and overnight, respectively. About 100 μL of the sample were injected on a Superose 6 30/100 (GE Healthcare) analytical SEC column preequilibrated with the last dialysis buffer. The UV absorbance at 280 and 260 nm were monitored to identify the different species.

### Hermes transposase stoichiometry determined by mass photometry (MP)[66]

The day before the experiment, three samples of frozen Hermes transposase (15% glycerol, 25 mM HEPES.Na pH 7.5, 750 mM NaCl, and 0.3 mM TCEP) were dialyzed for 2 h against buffer B500 (500 mM NaCl, 25 mM HEPES.Na pH 7.5, and 0.3 mM TCEP). One sample was left in B500 to equilibrate overnight, while the other two were transferred in B250 (250 mM NaCl, 25 mM HEPES.Na pH 7.5, and 0.3 mM TCEP) for 3 h. Finally, one sample was left overnight in B250, while the last sample was transferred to B150 (150 mM NaCl, 25 mM HEPES.Na pH 7.5, and 0.3 mM TCEP) for 16 h. A volume increase, as well as light precipitation, was observed as the NaCl concentration of the dialysis buffer decreased. The concentrations of the dialyzed samples were

10.0 μM (in B500), 6.5 μM (in B250), and 6.0 μM (in B150). Just before performing the MP experiments, the samples were diluted 10, 50, and 100 times in their respective buffers, and equilibrated for at least 10 min at room temperature. The detailed protocol was published by the NHLBI biophysics facility with the silicon gasket applied on a glass coverslip as a sample holder[67]. Data collection was performed on a G10 RefeynOne mass photometer. About 10 μL of buffer was used to optimize the focus, then 10 μL of the sample was mixed into the buffer drop for data collection (1 min acquisition). The concentrations that gave the best signal-to-noise were 50, 65, and 30 nM in B500, B250, and B150, respectively. The movie frames were processed with the built-in software DiscoverMP. The contrast values were converted to masses using a calibration curve obtained with BSA (monomer and dimer), alcohol dehydrogenase (monomer and dimer), ovalbumin, and thyroglobulin in a PBS buffer. The mass distributions were fitted using a Gaussian distribution model implemented into DiscoverMP. The theoretical mass of a Hermes monomer, dimer, hexamer, and octamer are 70.1, 140.2, 420.7, and 561.0 kDa, respectively.

### Transposase/DNA complexes stoichiometry determined by MP

A typical sample (100 μL) was composed of 10 μM Hermes mixed with 2.5, 5, or 10 μM DNA in 25 mM HEPES.Na pH 7.5, 750 mM NaCl, and 0.3 mM TCEP. The samples were successively dialyzed as described for the transposase alone to reach the equilibrium in B150. Just before the experiments, the samples were diluted 50 and 100 times and equilibrated for at least 10 min at room temperature. The same experimental procedure was used as described for the apoprotein. The composition of the protein/DNA complexes (P1 and P2 masses) were determined by subtracting the mass of either hexameric Hermes or octameric Hermes, and the remaining mass was then divided by the mass of the given DNAs. The theoretical MW of the Hermes hexamer and octamer are 420.7 and 561.0 kDa, respectively. The theoretical MWs of the LE-TIR, LE-TIR + 13, LE-TIR + 30, and 8 + LE-TIR + 30 DNAs are 10.1, 18.1, 32.2, and 33.2 kDa, respectively.

### Crystallization, X-ray diffraction data collection, structure determination, and model refinement of the BED/LE11-27 complex

After purification, BED was dialyzed overnight at 4 °C against 25 mM HEPES.Na pH 7.5, 100 mM NaCl, 0.3 mM TCEP and subsequently concentrated (Vivaspin 20 3 kDa MWCO, GE Healthcare) to ~10 mg/mL. The LE11-27 DNA was dialyzed against the same buffer and concentrated to ~2.5 mM (Vivaspin 500, 3 kDa MWCO, Merck). The complex was formed by mixing the protein and the DNA in a 2:1 protein-to-DNA ratio (final concentration of 800 and 400 μM, respectively). Crystals were grown at 20 °C by the hanging drop method. About 1.7 μL of the sample were mixed with 2.3 μL of crystallization solution composed of 100 mM Bis-Tris pH 6.5 (Hampton Research) and 25% PEG 4000 (Hampton Research). Crystals grew over 16 days to a size of ~0.3 × 0.3 × 0.3 mm and were cryoprotected by a quick transfer to a stabilizing solution at 20% ethylene glycol, 12.5 mM HEPES.Na pH 7.5, 50 mM NaCl, 25 mM Bis-Tris pH 6.5, 12.5% PEG 4000 prior to freezing in liquid nitrogen. The X-ray diffraction data were collected at the Advanced Photon Source beamline 22-ID, operated by SER-CAT on an Eiger X16M detector. Three anomalous diffraction data sets were collected on the same crystal around the zinc absorption edge, peak, and remote wavelengths (9661 eV and 1.28335 Å; 9665 eV and 1.28282 Å; 10000 eV and 1.23984 Å, respectively). The diffraction data were integrated and scaled with XDS and XSCALE[68]. Initial experimental electron density maps were computed with Autosol from the Phenix package[69]. The maps were improved by incorporating the anomalous signal of the DNA backbone's 8 P atoms and the 4 S atoms from the protein using an additional dataset collected at 8.03 keV using a rotating anode source equipped with an Eiger 4 M detector. Phase calculations were performed by using Sharp[70]. The density-modified

map enabled to build an initial model in COOT[71]. The "edge" diffraction data processed as a non-anomalous dataset (Friedel law considered as true) and treated for anisotropy (Ian J. Tickle, The STARANISO Server, http://staraniso.globalphasing.org/cgi-bin/staraniso.cgi) was used to further improve and refine the model in Phenix and Buster (Global Phasing Limited) at 2.5 Å resolution[69,72]. Detailed crystallographic statistics are in Supplementary Table 1.

## Cryo-EM specimen preparation

Frozen or freshly purified Hermes transposase was mixed in an 8:2 protein-to-DNA ratio with either 8 + LE-TIR + 30 or 8 + RE-TIR + 30 in a 250 μL sample (1 equivalent was equal to 5 μM). The samples were dialyzed against a series of buffers as described for the SEC interaction assay with the final buffer composed of 25 mM HEPES.Na pH 7.5, 150 mM NaCl, 0.3 mM TCEP, and protease inhibitors (Roche). The samples were concentrated to 100 μL and purified by SEC (Superose 6 30/100, GE Healthcare) at 4 °C. The fraction with the highest absorbance was concentrated to reach the absorbance of ~0.60 at 260 nm and ~0.5 at 280 nm. The protein concentration was estimated to be ~0.2 mg/mL by comparing the SDS-PAGE band intensity of the cryo-EM sample with that of a dilution series of apo-Hermes.

The samples were frozen on copper or gold grids with holey carbon film coated with a 2 nm continuous carbon film (Quantifoil R1.2/1.3 ultrathin carbon, 300 mesh) freshly glow discharged for 20 s at 15 mA (PELCO easiGlow). A Vitrobot Mark IV (FEI) rapid plunging device was used for the specimen preparation. The Vitrobot chamber was at room temperature and the relative humidity was set at 100%. Three μL of samples were applied on the grid and after 10 s wait, the excess sample was blotted for 3 or 4 s (force 1) and flash frozen in liquid ethane cooled by liquid nitrogen.

## Cryo-EM data collection

For the LE-LE transpososome, we collected ~9600 movies from two grids on a 200 kV Glacios TEM (FEI) equipped with a K3 direct electron detector camera (Gatan). The movies were recorded in super-resolution counting mode at a nominal magnification of 130,000, corresponding to a calibrated super-resolution pixel size of 0.58 Å per pixel with a defocus range from −1 to −2.5 μm. The acquisition was supervised by the semi-automated program SerialEM[73]. The dose rate on the camera was set at 15 electrons per physical pixel per second. The total exposure time for each movie was 2 s with a total exposure dose of 22.3 e−/Å² (1.39 e−/Å² per frame). Each movie was composed of 16 frames, with 125 ms per frame.

For the RE-RE transpososome, ~9500 movies were recorded on a 300 kV Titan Krios TEM (FEI) equipped with a K3 camera. The movies were recorded in super-resolution counting mode at a nominal magnification of 105,000, corresponding to a calibrated super-resolution pixel size of 0.43 Å per pixel with a defocus range from −1 to −2.5 μm. The acquisition was supervised by the semi-automated program SerialEM[73]. The dose rate on the camera was set at 22 electron per physical pixel per second. The total exposure time for each movie was 1.66 s with a total exposure dose of 48.7 e−/Å² (2.21 e−/Å² per frame). Each movie was composed of 22 frames, with 75 ms per frame.

## Cryo-EM single particle analysis

The single particle analyses were performed with RELION 3.1 ran on the NIH HPC Biowulf cluster (http://hpc.nih.gov)[74–76]. The processing of each dataset followed the same protocol unless stated differently. The processing workflows are presented in Supplementary Figs. 5, 6, 9. UCSF Chimera was used to visualize the maps[77]. The figures of cryo-EM map and models were prepared with Chimera or ChimeraX[77,78], and PyMol (http://www.pymol.org).

The movies were motion corrected in RELION 3.1 and binned by a factor 2, resulting in a pixel size of 1.16 Å (Glacios data) and 0.86 Å (Krios data), respectively, for further processing. The contrast transfer

function (CTF) parameters were estimated with gctf1.06[79]. The program crYOLO was used for particle picking with a general model on motion-corrected movies (~3.85 million and ~2.92 million initial particles for LE-LE and RE-RE transpososome, respectively)[80]. The best particles were selected by several rounds of 2D classifications. The particle stacks of the LE-LE transpososome were joined and ran through a last 2D classification. The resulting particle stack (~641,300 particles and ~186,493 particles for LE-LE and RE-RE transpososome, respectively) was used to generate a 3D initial model that was used as a reference for a 3D classification job (six classes). The best classes were used for gold-standard refinement (class #4 and #6 of the LE-LE transpososome and class #2 of the RE-RE transpososome in Supplementary Figs. 5, 9, respectively). For both samples, the equatorial Hermes dimers (B and D) were poorly resolved, and we opted for a multi-body refinement for the LE-LE transpososome and a partial signal subtraction strategy to improve the alignment of the core of the complex.

For the LE-LE transpososome a "consensus" gold-standard refinement was performed with the joined particles from 3D classes #4, #5, and #6 (~359,800 particles), followed by per-particle CTF refinement, Bayesian polishing, and a new 3D refinement with the shiny particles (4.89 Å resolution). The map was divided into three body masks. Mask Body 1 covered the DNAs, the Hermes dimers A and C, and the BED domains from Hermes dimer B bound to the DNAs. The masks Body 2 and 3 covered the N-truncated Hermes dimers B and D, respectively. These masks were used to run a multi-body refinement to obtain a composite map of the LE-LE transpososome. The Body 1 map was gold-standard refined at 4.64 Å resolution. The Body 2 and 3 maps could not be gold-standard refined but were estimated at 10.21 and 10.94 Å resolution, respectively.

For the RE-RE transpososome, the 3D class #2 was gold-standard refined to generate a mask that only covered the core of the transpososome, i.e., the DNAs and the Hermes dimers A and C, leaving out the equatorial Hermes dimers B and D. Partial signal subtraction was performed on the particle stack of class #2 (53,656 particles). The resulting "edited" particles were subjected to focused refinement with local orientational search. The resolution of the final map was 5.1 Å (0.143 FSC threshold).

## Model building - LE-LE transpososome

Two models of the N-truncated Hermes dimer bound to two nicked TIRs (PDB:6DX0)[9] and four BED/LE11-27 models (PDB: 8EB5, from this publication) were rigid-body fitted inside the Body 1 map (UCSF Chimera)[77]. The DNA base pairs and the BED domains that lay out of the density were removed. The contiguous pieces of DNAs were then merged as one dsDNA, and the sequence was corrected in COOT 0.9.6.2-pre EL[81] to generate two LE-DNAs. The BED domains of protomers A1, A2, C1, and C2 were linked to the N-truncated Hermes structures in COOT as well. B-form restraints were applied to the DNA base pairs and the all-atom real-space refinement against the Body 1 map was performed. The resulting model was further real-space refined against the cryo-EM map and validated in Phenix 1.19.2 (cryo-EM module). The refinement statistics are summarized in Supplementary Table 3. Two models of the apo N-truncated Hermes monomer (PDB: 2BW3)[10] were rigid-body fitted inside the Body 2 and Body 3 maps.

## Model building—core of the RE-RE transpososome

The model of the core of the RE-RE transpososome was generated in a similar fashion as the core of the LE-LE transpososome (Body 1), but the crystal structure of the N-truncated Hermes dimer bound to two cleaved TIRs (PDB: 4D1Q)[11] was also used as a starting point. The refinement statistics are summarized in Supplementary Table 3.

## BED/DNA electrophoretic mobility shift assay (EMSA)

The samples were prepared by mixing 0.9 μM of cold DNA in B150 (25 mM HEPES.Na pH 7.5, 150 mM NaCl, and 0.3 mM TCEP) with 0.1 μM

of 6FAM-labeled DNA in B150 (5′−6FAM top strand), 0.5 μM of ran17 DNA, various amount (0 to 10 μM) of BED protein in B150 and loading dye (0.5X TBE, 10% glycerol and bromophenol blue). The samples were equilibrated for 1 h at 4 °C, while the PAGE gels were pre-run at 80 V (15 % polyacrylamide, 0.5X TBE, 1 mm gels). The samples were equilibrated for 15 min at room temperature prior being loaded (10 μL) onto the gels. The PAGE was run for 4 h at 80 V in an ice box. The EMSAs were performed as technical triplicates. The gels were scanned with a Typhoon FLA7000 (GE Healthcare) fluorescence imager avoiding the saturation of the detector. We used ImageJ to extract the intensity of the delayed bands[82], and transformed as percentage of DNA bound. The means of the triplicate points were plotted against the concentration of BED and the standard deviation was reported as bars. The resulting binding curves were fitted in GraphPad Prism 9.3.1 with the equation: $Y = Bmax \times X^h/(Kd^h + X^h)$.

### In-cell transposition assay

The helper plasmid pFV4a-Hermes and the donor plasmids pHermesWT-CMVpuro, pHermes2LE-CMVpuro, pHermes2RE-CMVpuro, pHermes2LEmut-CMVpuro, pHermesLE50Scr-CMVpuro, pHermesLE140Scr-CMVpuro, and pHermes2LEmut2-CMVpuro were ordered from GenScript using the plasmids pFHelR and pHelR-CMV-puro from Grabundzija et al. (2018) as backbones[83]. The rest of the donor plasmids were obtained by deletion mutagenesis of pHermesWT-CMVpuro[84]. The sequences of the transposon ends are reported in Supplementary Table 4. The helper plasmid contains the gene of the Hermes transposase, and the donor plasmid carries the puromycin resistance gene flanked by the *Hermes* LE and RE and variants to form the puro[R] transposon.

HEK293T cells ($0.5 \times 10^6$) were seeded onto six-well plates 1 day before transfection. The cells were transfected with 1 μg of helper plasmid and 0.5 μg of the donor plasmid. All the transfections were performed with Lipofectamine 3000 (Thermo Fisher Scientific) according to the manufacturer's protocol. Two days post-transfection, the cells was replated onto 100 mm dishes at a 50-fold dilution (i.e., 2% of the cells) and selected for transposon integration with 2 μg/mL puromycin. The selection medium was changed every 3 days. After 8–12 days, the cell colonies were fixed directly into their dish for 20 min with 4% formaldehyde in PBS and subsequently stained overnight with 1% methylene blue in PBS for counting. All experiments were independently replicated at least four times. Negative control experiments were performed by replacing the helper plasmid with pUC19. The *piggyBac* system was used as a procedure control.

All statistical analyses were performed using GraphPad Prism 9.3.1. Two-tailed Student's unpaired *t*-tests were used to compare means between experimental samples and their corresponding pUC19 controls and means between some experimental samples and pHermesWT-CMVpuro. The bars correspond to the standard deviations.

### Reporting summary

Further information on research design is available in the Nature Portfolio Reporting Summary linked to this article.

## Data availability

The structure factors and the crystal structure of the Hermes transposase BED domain bound to the Hermes transposon LE quasi-palindrome LE-STR1-STR2 have been deposited in the Protein Data Bank (PDB) under the accession code 8EB5. The cryo-EM maps of the core of the LE-LE and of the RE-RE *Hermes* transpososomes were deposited in the Electron Microscopy Data Bank (EMDB) under the accession codes 28034 and 40553, respectively. Their related structure models were deposited in the PDB with the codes 8EDG and 8SJD, respectively.

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

## Acknowledgements

This work was supported by the Intramural Program of the National Institute of Diabetes and Digestive and Kidney Diseases, National Institutes of Health (NIDDK/NIH) under Grant DK036153-16. X-ray diffraction data were collected at the Southeast Regional Collaborative Access Team (SER-CAT) 22-ID beamline at the Advanced Photon Source (APS), Argonne National Laboratory. SER-CAT is supported by its member institutions (https://www2.ser.aps.anl.gov/about_us.html#TITLE_Memberships), and equipment grants (S10_RR25528 and S10_RR028976 and S10_OD027000) from the NIH. The use of the Advanced Photon Source was supported by the US Department of Energy, Office of Science, Office of Basic Energy Sciences, under Contract No. W-31-109-Eng-38. The computations for the cryo-EM single particle analysis were carried out using the High-Performance Computing Systems at the NIH. The authors thank Greg Piszczek and Di Wu of the Biophysics Core Facility at the National Heart, Lung, and Blood Institute (NHLBI/NIH) for the access to the mass photometer. The authors would like to also thank Huaibin Wang and Yanxiang Cui for their help with cryo-EM data collection at the Multi Institute Cryo-EM Facility (MICEF), NIDDK/NIH. The authors are also thankful to Istvan Botos (NIDDK/NIH) for on-site computational support.

## Author contributions

L.L. performed most of the experiments in the study except for the in-cell transposition assays on the truncated LE that were carried out by C.M.F. L.L. analyzed all the data generated in the study. L.L. and F.D. analyzed the X-ray data. L.L. prepared all the samples used in this study, including the crystals and the cryo-EM specimens. A.B.H. initiated the project. F.D. supervised the study. All authors participated in manuscript preparation.

## Funding

## Competing interests

The authors declare no competing interests.
