## [Peer Review File · Nature Communications]

Zinc-finger BED domains drive the formation of the active Hermes transpososome by asymmetric DNA bindingREVIEWER COMMENTS

Reviewer #1 (Remarks to the Author):

Lannes et al., report the crystal structure of DNA bound BED domain of Hermes transposon as well as cryo-EM structures of the full-length Hermes transposon bound with either LE-DNA or RE-DNA. While multiple structures of truncated Hermes transposon have been reported previously, the full-length Hermes transposon structure described here reveals several new interesting features. The most interesting discovery is that the BED domain could bind to the LE-DNA but not RE-DNA. I think this work advances our understanding of the role of BED domain in Hermes transposon mediated transposition. However, the structural work presented here requires some improvement before it could be published.

(1) Overall, the cryo-EM works are of relatively low quality. The cryo-EM maps look noisy and featureless. It seems to me that the major issue is that the particles used for 3D reconstruction have a certain degree of preferential orientation, as evidenced by Figure S6 and S8. The particles preferential orientations could result in low resolution reconstruction and isotropic cryo-EM map, which would affect the precise model modelling and structural interpretations. My sense is that the overall fitting of DNA and Hermes transposon is accurate, as the crystal structures of the majority part of Hermes transposon have been determined by previous and current work. However, the structural artifacts caused by the resolution anisotropy would still compromise the accuracy of model building. To increase the impact of this work, I would encourage the authors to improve the quality of the cryo-EM maps. The issue of the preferential orientation could be overcome by several established methods, such as tilting the compustage of EM or depositing a suitable substrate (graphene/continuous carbon) on top of the grids.

(2) The authors need to show the fitting of DNA and each domain of Hermes transposon into the cryo-EM densities in at least two perpendicular views to convince the readers that the modeling is reliable. The α -helix and β -sheet should be well resolved in the 4-5 Å cryo-EM map. The representative densities of α -helix and β -sheet for each cryo-EM map should be shown in supplementary figures.

(3) The FSC curves between models and maps should be shown in supplementary figures.

Reviewer #2 (Remarks to the Author):

The manuscript by Lannes et al. describes the structural and functional characterization of the Hermes BED domains and the BED-binding signatures in the left and right ends of the transposon. The work is of excellent technical quality, and the cryo-EM structures partly explain Hermes' complex architectural organization. The binding of Hermes to LE-LE DNA involves the BED domains from three Hermes dimers in the complex. Conversely, binding to RE-RE DNA does not involve any of the BED domains of the octamer. This highlights the complexity and asymmetry of the Hermes transpososome and elevates the significance of this work. The manuscript does not entirely resolve the contradiction as to why the Hermes dimer is competent for all functional activities *in vitro* but needs a much more complex architecture for transposition *in vivo*. However, it significantly advances our limited understanding of hAT transposases.

Some sections of the manuscript are difficult to follow and not well justified. My comments below refer to areas that could be addressed to improve the flow of the manuscript and support to proposed transpososome model.

The description of the crystal structure needs to be clarified. Since the DNA is not a perfect palindrome and the DNA duplex forms through crystallographic symmetry, can the authors unequivocally assign the DNA sequence in the electron density maps? The authors describe the specific protein:DNA interactions and how the alternate conformations of the N67/R70 pair allow the interaction with the imperfect palindrome. Showing the quality of the electron density and explaining how the DNA sequence was assigned is necessary to describe these interactions and their functional implications.

The authors use an elegant combination of SEC and MP to assemble the transpososome for structural characterization, leading to the reconstruction of the Hermes octamer bound to two left or two right ends. The authors state that the equatorial dimers are flexible and exclude them from refinement and reconstruction. However, they use preliminary maps to determine the connectivity of the additional BED domains contributing to DNA binding in the LE-LE transpososome. Excluding ~50% of the protein, complement seems a drastic approach, mainly when a standard processing protocol was used. I wonder if a different processing approach may have given more information. For instance, did the authors try multi-body refinement (Relion), non-uniform refinement (cryosparc) or masked refinement to improve the connectivity of the density in the flexible regions? These are robust approaches optimized for flexible samples and could have improved the quality of the final reconstructions.

In the initial 3D reconstruction (as well as classes #3 and #4), one of the equatorial dimers seems less defined than the other. Is the data set a mixture of trimers of dimers and tetramers of dimers? And could that explain why only three dimers contribute BED domains to the recognition of the LE? Alternatively, could the LE-RE transpososome be a trimer of dimers where the present equatorial dimer engages the LE, and the "absent" equatorial dimer would allow binding to the RE? These may not be reasonable explanations, but is it possible that the sole role of the equatorial dimers is for one of them to provide one single BED domain to the complex?

Class #5 is the major class after 3D classification, but this class was not refined further. Why?

Minor comments:

The last two sentences of the abstract are repetitive and may not be needed.

Line 60 formatting error.

Reviewer #3 (Remarks to the Author):

In this manuscript, Lannes and colleagues seek to understand how Hermes carries out transposition in cells, where both the protein and substrate requirements differ from what is required in vitro. Here, they focus on the role of a zinc finger BED domain in Hermes and how it influences assembly of the transpososome. I enjoyed reading this paper and think it represents an important contribution to understanding these mobile elements. My only overall concern is that work still leaves the reader with an in vitro view of the transposition mechanism; the requirement for ~200 bp LE and RE sequences in cells is not understood.

Main questions/comments

For DNA-binding experiments in Fig 2 and then again in Fig S9, why is SEC used? The results are subtle and difficult to quantitate. Fluorescence polarization would have been easy for these experiments and would have provided affinity data, the lack of which is a weakness in the paper.

Fig 3E. These results are very difficult to follow. SEC in B looks very similar to E. Figs S4 and S5 are much more revealing with the mass photometry data. Those panels should be in the main figure and the SEC alone could be in supplemental.

Fig 4 and related. Why remove the equatorial protomers in the structure? Just because they are weaker density? They are clearly there, at least in one 3d class, and the BED domains are present. The domains could easily be removed for drawings to improve clarity, but to just not fit them and include them in the structure seems odd. The authors should state that they could not be fit/interpreted if this is the case.

What is the structural/biochemical basis for the AT-rich sequence requirement flanking the core

STR sequence? The authors show that it is important, but do not explain it.

The cooperative nature of binding in Fig 6 is very interesting and the conclusion that it comes from DNA distortion makes sense. However, it is difficult to compare affinities because there are no error estimates. Replicates and standard deviations are needed here. Are the authors proposing cooperative, but 2-fold weaker binding for the palindromic STR?

In Fig 8 and related discussion, the proposal that the higher affinity LE complex forms first and then recruits the weaker RE makes good sense and is supported by the new data. However, what is the role of the remaining DNA in LE and RE that makes it so important in cells? I found it surprising that the deletion experiments were done to identify ~200 bp as the minimum length, but that the STR sequences weren't altered to determine if they are needed. A role for additional Hermes octamers bridging LE and RE was not discussed as a possible explanation.

Minor things

Fig 2d. The S66-A15N7 interaction shown for the BED1 interface looks like it must be quite long. Is this really a hydrogen bond?

Fig 2. Are any STRs panindromic with respect to the optimal N67 and R70 interactions observed for the sequence chosen?

The authors should say how many nucleotides are between cleavage sites in the target (intro and/or Fig 1)

What zn fingers do the BED domains most closely resemble on a structural level (DALI)?

The Hermes octamer forms a ruler that defines (roughly) the LE and RE sequences that are captured by the non-catalytic C dimer. How does this sequence compare to the TIR sequence and how is Hermes prevented from cleaving them?

The authors should explain earlier in the paper that parallel orientation of LE/RE is expected in the transpososome and explain why; don't wait for the discussion. Many general readers may miss this in Fig 4.

Line 259. Bridged by two dna duplexes (not oligos)

Line 336. Need a better section title. Readout is transposition, not formation of synaptic complex. Also, wording is difficult to follow.

Line 411 - why is only binding to RE-TIR important? Doesn't binding by dimer C make a contribution as well?

Discussion - is a key role of nucleosomes likely in this case? The experimental readout is using pUC-like plasmids, which may not even have nucleosomes when Hermes synaptosomes form in HEK293 cells.

POINT-BY-POINT RESPONSE TO REVIEWER COMMENTS

Reviewer #1:

Lannes et al., report the crystal structure of DNA bound BED domain of Hermes transposon as well as cryo-EM structures of the full-length Hermes transposon bound with either LE-DNA or RE-DNA. While multiple structures of truncated Hermes transposon have been reported previously, the full-length Hermes transposon structure described here reveals several new interesting features. The most interesting discovery is that the BED domain could bind to the LE-DNA but not RE-DNA. I think this work advances our understanding of the role of BED domain in Hermes transposon mediated transposition. However, the structural work presented here requires some improvement before it could be published.

(1) Overall, the cryo-EM works are of relatively low quality. The cryo-EM maps look noisy, and featureless. It seems to me that the major issue is that the particles used for 3D reconstruction have a certain degree of preferential orientation, as evidenced by Figure S6 and S8. The particles preferential orientations could result in low resolution reconstruction and isotropic cryo-EM map, which would affect the precise model modelling and structural interpretations. My sense is that the overall fitting of DNA and Hermes transposon is accurate, as the crystal structures of the majority part of Hermes transposon have been determined by previous and current work. However, the structural artifacts caused by the resolution anisotropy would still compromise the accuracy of model building. To increase the impact of this work, I would encourage the authors to improve the quality of the cryo-EM maps. The issue of the preferential orientation could be overcome by several established methods, such as tilting the compustage of EM or depositing a suitable substrate (graphene/continuous carbon) on top of the grids.

We agree with the reviewer that our maps have a rather low resolution. The Hermes transposase complexes suffer from denaturation at the air/water interface that we were only able to avoid by using EM grids coated with a continuous carbon film (see Material and Methods). As noted by the reviewer, another way to prepare samples would be to use graphene oxide (GO), and indeed we also tried GO coating. In fact, we invested considerable effort attempting to use GO and screening grids on a 200kV scope; unfortunately, the results were not any better and, in our hands, we had better results with the continuous carbon film. Another problem that compromises the data quality was that the transpososomes were more stable when embedded in thick ice. This gave rise to higher-than-ideal background because of scattering from the ice. Together, these experimental constraints - the carbon film and the rather thick ice - led to the significant absorption of electrons and consequently limited the final resolution.

In addition, what appears to be one of our key findings is that the equatorial Hermes dimers do not interact tightly with the DNA. Thus, they are somewhat mobile relative to the top and bottom dimers which results in poor particle alignment. As a consequence, regions of the maps corresponding to the equatorial Hermes dimers are poorly resolved with broken density. In order to obtain better densities for the B and D equatorial dimers, we attempted partial signal subtraction by masking out and subtracting the signal of the A and C dimers (i.e., top and bottom). Unfortunately, these efforts didn't yield interpretable densities for the equatorial dimers, presumably due to their inherent mobility, although interpretable densities were possible to obtain for the two BED domains of the dimer B that bound to STR2 in the LE-LE transpososome.

We agree with the reviewer suggestion that a "certain degree of preferential orientation could result in low resolution reconstruction and isotropic cryo-EM map". Due to the ring shape of the

complex, the 2D classes gave the impression that only two views (90-degree from each another) were present. While the orientation distribution was not ideal, the Euler angular distribution of the particles that contributed to the final map of the LE-LE transpososome showed that we had full angular coverage of the map. This was certainly in contrast to that of the RE-RE transpososome where some orientations were missing (compare Figures S6 and S9). We note however that the particles that contributed to an earlier RE-RE map had full angular coverage. After the subtraction of the signal of the B and D equatorial dimers and subsequent 3D classification, the remaining particles indeed had missing orientations. Despite this, the map did not show features of a severe preferred orientation (i.e. streaking of the map), or features that would indicate overfitting (bumpy global FSC curves with a short drop, global resolution much higher than the map-to-model resolution as illustrated in Tan et al. Nature Methods, 2017).

Nevertheless, as we agreed with the reviewer that we should attempt to improve on the orientation distribution, we have now collected another Krios data set at 300kV for the RE-RE transpososome with a stage tilt set at 30 degrees. While in this data set the orientation distribution has improved (see figure below, A), unfortunately this did not yield better map resolution or features, perhaps due to the additional ice length the electron beam had to pass through (compare B and C in figure below). Even at this low resolution, the map once again clearly supports the presence of two DNA molecules and there is some density for the equatorial dimers.

A. Processing of the 30°-tilt data of the RE-RE transpososome

B. 0°-tilt data of the RE-RE transpososome at a similar processing stage as the 30°-tilted data refined map above

C. (0°-tilt data) Final map of the core of the RE-RE transpososome

The reviewer commented that the maps shown in the manuscript looked noisy and featureless. We apologize for this. It was the consequence of our choice to show the maps at a low-density level in order to bring up the less-resolved, low signal regions of the map. Unfortunately, this

increased the noise of the maps. In an attempt to find a better balance between the quality of the presented maps while also being able to show regions of lower signals and to improve the signal-to-noise of the maps, we have used the DeepEMhancer post-processing algorithm (Sanchez-Garcia et al., Communications Biology 2021) inputting the two half maps of the final RELION refinement. This algorithm performs sharpening and denoising. Below is shown a side-by-side comparison for the maps of the cores of the LE-LE and RE-RE transpososomes. The DeepEMhancer results are less noisy, the protein chains and the DNA strands are more visible, and it is easier to appreciate the features of the maps.

The denoised maps are now presented in Fig. 4 and Fig. 5. (For clarity, we have also now separately colored each Hermes monomer and DNA strand.)

(2) The authors need to show the fitting of DNA and each domain of Hermes transposon into the cryo-EM densities in at least two perpendicular views to convince the readers that the modeling is reliable. The α -helix and β -sheet should be well resolved in the 4-5 Å cryo-EM map. The

representative densities of a-helix and b-sheet for each cryo-EM map should be shown in supplementary figures.

We agree and thank the reviewer for this suggestion. New Figure 4E now shows the model of the LE-LE transpososome inside the multi-body composite map. New Figure S8 now shows the fitting of the core of the LE-LE transpososome with the details of each LE-DNA and each Hermes domain as well as representative density for two a-helices of the intertwined domain and the five-stranded b-sheet of the RNase H-like catalytic domain. Figure 5 and new Figure S10 show the fitting of the model of the core of the RE-RE transpososome in its cryo-EM map.

(3) The FSC curves between models and maps should be shown in supplementary figures.

Again, the referee is correct and the FSC curves are now included in new Figure S7 and Figure S10.

Reviewer #2:

The manuscript by Lannes et al. describes the structural and functional characterization of the Hermes BED domains and the BED-binding signatures in the left and right ends of the transposon. The work is of excellent technical quality, and the cryo-EM structures partly explain Hermes' complex architectural organization. The binding of Hermes to LE-LE DNA involves the BED domains from three Hermes dimers in the complex. Conversely, binding to RE-RE DNA does not involve any of the BED domains of the octamer. This highlights the complexity and asymmetry of the Hermes transpososome and elevates the significance of this work. The manuscript does not entirely resolve the contradiction as to why the Hermes dimer is competent for all functional activities in vitro but needs a much more complex architecture for transposition in vivo. However, it significantly advances our limited understanding of hAT transposases. Some sections of the manuscript are difficult to follow and not well justified. My comments below refer to areas that could be addressed to improve the flow of the manuscript and support to proposed transpososome model.

(1) The description of the crystal structure needs to be clarified. Since the DNA is not a perfect palindrome and the DNA duplex forms through crystallographic symmetry, can the authors unequivocally assign the DNA sequence in the electron density maps?

We thank the reviewer for raising this issue and we agree that the original description of the crystallography was not sufficient. The situation we have encountered is not common but it is also not a rarity. For example, it happens fairly often in the case of HIV protease crystal structures where a crystallographic two-fold is also the molecular axis, yet a peptidomimic inhibitor lacking such symmetry still spans the two-fold-symmetric active site. The resulting electron density is therefore the superposition of the two two-fold related inhibitor densities. Our situation is analogous. There are cases when the situation can be improved by integrating and scaling the X-ray data in a lower symmetry space group. This helps if the two-fold axis intersecting the DNA is in fact a non-crystallographic axis that is nearly crystallographic and there are some small but detectable differences between parts of the assembly related by this axis.

Therefore, we have re-processed the X-ray data in space group P6(5) (rather than in the apparent space group P6(5)22) and refined the structure. As expected, the data processing statistics were very similar to what we obtained in P6(5)22. In the P6(5) space group, the symmetric unit contains two now crystallographically independent BED domain monomers and the full dsDNA duplex. However, as the orientation of the dsDNA oligo is not clear, as the

density showed what appeared to be the superposition of the two orientations, we performed two additional independent X-ray refinements with the two possible orientations. If one of the orientations were correct and the other not, the expectation is that the R_{free} for the correct orientation would be lower than for the incorrect one. The resulting electron densities still showed indications of alternate conformations as before, and DNA base densities were also consistent with two base possibilities being simultaneously present. The numerical results are as follows: The two R_{free}s were 33.6% and 33.7% respectively for the two orientations, while the working R factors were 26.7% and 26.3%. We do not believe that the difference between these two numbers is significant, and they are all higher than the R_{free}/R_{work} numbers at 23.8%, 23.1% respectively that we obtained in space group P6(5)22. Therefore it does not appear that the crystal lattice favored one orientation vs the other, and we have elected to report the structure in P6(5)22.

To answer the reviewer's question directly, the assignment of the sequence to the DNA was not problematic as the two blunt ends of the dsDNA oligo were clear as they showed up as breaks in the backbone chains. As the locations of the base densities were clear, we had no problem in establishing the correct register. However, it is certainly true that at those positions that deviate from a perfect palindrome, the density is a superposition of the two possibilities. We believe that this indicates that the complex, composed of two BED domains and the dsDNA duplex, can occupy the crystal lattice positions in two possible orientations and the lattice does not select one over the other, so there is a statistical mixture throughout the crystal. This appears to be consistent with the fact that the lattice contacts are at the blunt ends of the dsDNA oligo, making both orientations equally possible.

To reflect all of the above, we have extensively rewritten lines 126-137 where we describe the crystallographic results.

(2) The authors describe the specific protein:DNA interactions and how the alternate conformations of the N67/R70 pair allow the interaction with the imperfect palindrome. Showing the quality of the electron density and explaining how the DNA sequence was assigned is necessary to describe these interactions and their functional implications.

We agree with the reviewer that electron density in the regions showing the alternate conformations should be included. It is now new Figure S2A. Regarding the DNA sequence assignment, please see the previous comment.

(3) The authors use an elegant combination of SEC and MP to assemble the transpososome for structural characterization, leading to the reconstruction of the Hermes octamer bound to two left or two right ends. The authors state that the equatorial dimers are flexible and exclude them from refinement and reconstruction. However, they use preliminary maps to determine the connectivity of the additional BED domains contributing to DNA binding in the LE-LE transpososome. Excluding ~50% of the protein, complement seems a drastic approach, mainly when a standard processing protocol was used. I wonder if a different processing approach may have given more information. For instance, did the authors try multi-body refinement (Relion), non-uniform refinement (cryosparc) or masked refinement to improve the connectivity of the density in the flexible regions? These are robust approaches optimized for flexible samples and could have improved the quality of the final reconstructions.

We thank the reviewer for the suggestions for improving the quality of the final reconstructions. We have tried all three of them. We used the particles from 3D classes #5 and #6 and performed masked 3D refinement with partial signal subtraction keeping the equatorial dimers

and Cryosparc's non-uniform refinement. We used the particles from 3D classes #4, #5 and #6 and performed multi-body refinement in RELION with the transpososome divided into three bodies. Below are shown the maps obtained for each of the processing strategies that can be compared to the "normal" 3D refinement.

A. "Normal" 3D refinement (Relion)

B. Multi-body refinement (Relion)

C. Non-uniform refinement (CryoSPARC)

D. Masked refinement with partial signal subtraction (Relion)

As the images indicate, the quality of multi-body refinements in RELION were indeed superior to the previous full particle refinements and therefore we have replaced the relevant figures. The multi-body refinement now shows acceptable densities for the equatorial dimers, and importantly we do not have to mask them now out in order to be able to fit the molecular model.

(4) In the initial 3D reconstruction (as well as classes #3 and #4), one of the equatorial dimers seems less defined than the other. Is the data set a mixture of trimers of dimers and tetramers

of dimers? And could that explain why only three dimers contribute BED domains to the recognition of the LE?

Yes, the reviewer is correct: based on our MP and SEC data, it is likely that our sample is a mixture of Hermes hexamer and octamer, and this is represented in the two different reconstructions shown in Fig. 4B and 4C. The presence of an initial 3D class (#4) that completely misses one of the equatorial Hermes dimers supports the presence of the hexamer. However, the particles grouped in the "hexamer" classes are less abundant (19%; Fig S5) than the particles grouped in the "octamer" classes (37%; Fig. S5), and we conclude that the Hermes hexamer is not the most abundant oligomer.

In our maps, the two equatorial BED domains bound to the DNAs face towards the equatorial dimer B. In the case of the hexameric Hermes complex, these BED domains can only come from the equatorial dimer B as it is the only one present. In the case of the octameric Hermes complex, there are only two equatorial BED domains and they similarly face dimer B. It would be reasonable to propose that this dimer is the only provider of equatorial BED domains. This hypothesis is supported by weak density that links the core of the dimer B with the BED domain interacting with the STR2 of LE-DNA 1. Due to the lack of density connecting the other equatorial BED to the core of a Hermes dimer, we cannot rule out that it belongs to dimer D.

There is no hint of a fourth BED domain binding to either of the single DNA ends (which could only occur between the STR2 and STR3) in any of the octamer classes, not even at the lowest-density level. Thus, our data suggests that inside the octamer complex, overall there are only 6 BED domains bound to the DNAs.

Alternatively, could the LE-RE transpososome be a trimer of dimers where the present equatorial dimer engages the LE, and the "absent" equatorial dimer would allow binding to the RE?

This is an interesting possibility; however, we do not have data supporting it. We have never observed a Hermes dimer interacting with only one end. The only evidence relating to dimer activity is the in vitro data which indicates that a dimer in principle is able to synapse two ends and is capable of carrying out the chemical steps necessary for transposition, but only at low and not physiologically relevant ionic strength. We therefore prefer not to speculate beyond what we have presented in Figure 8.

These may not be reasonable explanations, but is it possible that the sole role of the equatorial dimers is for one of them to provide one single BED domain to the complex?

In addition to any role in stabilizing the overall complex, we indeed conclude that the main role of the equatorial dimers is to provide a BED domain to the complex. We have added a sentence stating this to the Discussion (lines 455-458), as clearly in our previous version this was not explained well.

(5) Class #5 is the major class after 3D classification, but this class was not refined further. Why?

We thank the reviewer for pointing this out and we apologize as this was a labeling mistake on our part. The percentages of classes #5 and #6 had been erroneously swapped. Both classes have been refined as can be seen in Fig. S5 which shows how different particle stacks led to maps with different resolutions.

The last two sentences of the abstract are repetitive and may not be needed.

We agree with the reviewer and have deleted the final sentence from the abstract.

Line 60 formatting error.

We again thank the reviewer for pointing out this error. The references were not formatted properly, and they have now been corrected.

Reviewer #3:

In this manuscript, Lannes and colleagues seek to understand how Hermes carries out transposition in cells, where both the protein and substrate requirements differ from what is required in vitro. Here, they focus on the role of a zinc finger BED domain in Hermes and how it influences assembly of the transpososome. I enjoyed reading this paper and think it represents an important contribution to understanding these mobile elements. My only overall concern is that work still leaves the reader with an in vitro view of the transposition mechanism; the requirement for ~200 bp LE and RE sequences in cells is not understood.

We share the concern of the reviewer and have since performed additional experiments in cells that provide more information than was available upon our original submission, as detailed below. Our goal was to understand the reason for the very unusual assembly that Hermes forms and resolve the discrepancy between in vitro and in cell data. We would of course like to understand the requirement for the very long ends (which is not uncommon in eukaryotic DNA transposition) but it may involve other cellular factors, the exploration of which is beyond the scope of the current work.

Main questions/comments

(1) For DNA-binding experiments in Fig 2 and then again in Fig S9, why is SEC used? The results are subtle and difficult to quantitate. Fluorescence polarization would have been easy for these experiments and would have provided affinity data, the lack of which is a weakness in the paper.

In the experiments shown in Fig. 2, we were seeking evidence for site-specific binding of LE DNA and to determine the binding stoichiometry. As our eventual goal was to prepare grids for cryo-EM from solution, we used SEC as it is a well-established method to evaluate the stability of macromolecular complexes to get a qualitative sense of the binding affinity of the protein and its DNA partners. The successful isolation of the complex in the timescale of the chromatographic run ensures that the complex is enough stable for structural experiments. For these reasons, we chose to use SEC over other biophysical methods.

In old Fig. S9 (now new Fig. S11), we tested the capacity of several mutated DNA sequences to form stable complexes with the BED domain. Using SEC, we could rapidly estimate whether the various mutated sequences bound as well as LE11-27 (along with a measure of their binding stoichiometry) or with binding properties closer to that of ran17.

However, we agree with the reviewer that SEC is not suitable to obtain quantitative results. Therefore, following the advice of the reviewer, we measured the fluorescence polarization (FP) of FAM-labelled ran17, LE11-27, and LE11-27mut (at 5 nM and 50 nM) in presence of increasing amount of BED (50 nM – 50 μ M, 12 titration points in triplicate). The data for [DNA] = 50 nM is presented below with the change of FP plotted against BED concentration on a

logarithmic axis. The three datasets do not form a sigmoidal binding curve as expected; rather, we observed an increase of the change in FP that never reach a plateau even at high excess of BED. These measurements suggested poor binding affinity ($K_d \geq 50 \mu\text{M}$) and were inconsistent with our SEC and EMSA data. After consultation with colleagues who are experts in the technique, we concluded that our system has hit the limit of this method as the molecular weights (MW) of the binding partners are unfortunately very close to each other, with $\text{MW}(\text{LE11-27}) = 10.38 \text{ kDa}$ and $\text{MW}(\text{BED}) = 9.13 \text{ kDa}$. It seems that a BED domain (or even a pair of BED domains) is not large enough to significantly change the FP of the labelled DNA upon binding.

We have instead returned to EMSAs and attempted to measure binding affinity using this approach (see below in response to point 5).

(2) Fig 3E. These results are very difficult to follow. SEC in B looks very similar to E. Figs S4 and S5 are much more revealing with the mass photometry data. Those panels should be in the main figure and the SEC alone could be in supplemental.

We agree with the reviewer, and have merged Figures S4 and S5 to make a new Figure 3. The old Figure 3 is now new Figure S4.

(3) Fig 4 and related. Why remove the equatorial protomers in the structure? Just because they are weaker density? They are clearly there, at least in one 3d class, and the BED domains are present. The domains could easily be removed for drawings to improve clarity, but to just not fit them and include them in the structure seems odd. The authors should state that they could not be fit/interpreted if this is the case.

We removed the equatorial Hermes dimers, with the exception of two of their BED domains, as given their mobility they compromised particle alignment during 3D refinement. However, following reviewer 2's suggestion, we have now tried a number of other refinement methods beyond simple masking out the equatorial dimers. As detailed in our response to reviewer 2, RELION's multi-body refinement gave the best results for the LE-LE transpososome.

(4) What is the structural/biochemical basis for the AT-rich sequence requirement flanking the core STR sequence? The authors show that it is important, but do not explain it.

It has been shown that the affinity of transcription factors (TF) - and notably zinc-finger TFs - for their cognate binding site can be modulated by the content of the flanking sequences (Yella and Bhimsaria et al., Flexibility and structure of flanking DNA impact transcription factor affinity for its core motif, NAR, 2018). We observed that the 5'-AAGT motif flanked by an AT-rich sequence in its 3'-flank was more conducive for BED binding compared to when it was flanked by a sequence with a higher GC-content. In our crystal structure of the BED/DNA complex, three lysines of the BED domains interact with the phosphate groups of three nucleobases positioned at the 3'-end of specific binding sites. This interaction seems to be facilitated by the bending of the DNA. As AT-rich sequences tend to be intrinsically bent, in the case of an AAGT motif

followed by an AT-rich sequence, it is plausible that the intrinsic bending of the 3'-end of the motif favors the interaction of these three lysines thus overall stabilizing the BED binding. We have now added two sentences to summarize this possibility to the Discussion (lines 471-476).

(5) The cooperative nature of binding in Fig 6 is very interesting and the conclusion that it comes from DNA distortion makes sense. However, it is difficult to compare affinities because there are no error estimates. Replicates and standard deviations are needed here.

The reviewer is correct that replicates are needed to assess the internal variability of the EMSA experiments. Therefore we have re-run the same experiments (ran17 or LE11-27mut or LE11-27 DNAs vs BED) but in triplicate. (The uncropped original pictures are presented in Fig. S12 and the edited picture of one of each EMSA is showed in the new Fig. 6B). The binding curves are presented in the new Fig. 6C (also copied below), and we chose to plot them together to make it easier to compare their affinities. The standard deviation is reported for each titration point as bars centered around the mean value. Fig. 6C also reports the K_d values for each DNA within the 95% confidence interval. We believe that our binding curves are now more reliable and make our conclusions stronger.

(6) Are the authors proposing cooperative, but 2-fold weaker binding for the palindromic STR?

Yes, this is what the binding data in Fig. 6 and Fig S11 show. We observed that (1) the 5'-AAGC motif cannot form a stable complex with the BED domain regardless of the GC-content of its flanking sequences, and (2) the 5'-AAGT motif tightly bound to the BED domain only if it is flanked by an AT-rich sequence at its 3'-end (see Fig. S11). Based on these observations one would expect that LE11-27, with both of these motifs in a quasi-palindromic organization (5'-AAGTGGCTT-3'), would not be a tight binder. However, it is not the case, and not only one but two BED domains bind the quasi-palindrome with a K_d of 2.5 μM. Therefore, the most parsimonious explanation is that two weak motifs leverage each other's binding to create two strong binders.

(7) In Fig 8 and related discussion, the proposal that the higher affinity LE complex forms first and then recruits the weaker RE makes good sense and is supported by the new data. However, what is the role of the remaining DNA in LE and RE that makes it so important in cells? I found it surprising that the deletion experiments were done to identify ~200 bp as the minimum length, but that the STR sequences weren't altered to determine if they are needed. A role for additional Hermes octamers bridging LE and RE was not discussed as a possible explanation.

The reviewer is correct that altering the inner STR sequences was the next obvious experiment to investigate why longer DNA is needed in cells. We have now performed the suggested experiment by mutating the putative BED binding sites inside the transposon LE. The results are reported in new Fig. 7E, and described in the new section entitled "The binding of auxiliary Hermes transposases inside the transposon ends to support transposition in cells is unlikely".

We observed some decrease in transposition when the STR sequences beyond LE50 were mutated, but interestingly scrambling the sequence beyond LE50 and eliminating all STRs reduced transposition to baseline. While we are very interested to investigate this issue further, this would be beyond the scope of the current work.

Minor things

(8) Fig 2d. The S66-A15N7 interaction shown for the BED1 interface looks like it must be quite long. Is this really a hydrogen bond?

Following the reviewer's question, we have closely looked at this interaction and find that the distance between the atoms (3.3 Å) is in the range that consistent with a hydrogen bond.

(9) Fig 2. Are any STRs palindromic with respect to the optimal N67 and R70 interactions observed for the sequence chosen?

Yes, there is only one other location in the transposon ends that has a palindromic arrangement of STRs, and it is in the LE from bp 283 to 291. The STRs are highlighted in the sequence of the transposon ends in Fig. S3. The sequence here is identical to STR1-STR2.

(10) The authors should say how many nucleotides are between cleavage sites in the target (intro and/or Fig 1).

The reviewer is correct that this should have been stated. It is now included in both Figure 1 and in the third paragraph of the introduction.

(11) What zn fingers do the BED domains most closely resemble on a structural level (DALI)?

We thank the reviewer for the question. DALI produced only a short list of structural homologs (the complete list of matches against the full PDB is copied below), only one of which is a Zn-binding domain. This is another BED domain (PDB: 2DJR NMR structure) belonging to the zinc finger BED domain-containing protein 2 (ZBED2). Both structures present a similar folding pattern except that the structure of ZBED2 does not have the alpha-helix 1, possibly because the N-terminal sequence was truncated.

No: Chain Description

- 1: 6il9-A MOLECULE: FRUCTURONATE-TAGATURONATE EPIMERASE UXAE FROM COH
- 2: 1nrj-A MOLECULE: SIGNAL RECOGNITION PARTICLE RECEPTOR ALPHA SUBUNI
- 3: 6c3k-B MOLECULE: PENICILLIN-BINDING PROTEIN 4;
- 4: 6ilb-D MOLECULE: FRUCTURONATE-TAGATURONATE EPIMERASE UXAE;
- 5: 5yud-A MOLECULE: BACULOVIRAL IAP REPEAT-CONTAINING PROTEIN 1E;
- 6: 2djr-A MOLECULE: ZINC FINGER BED DOMAIN-CONTAINING PROTEIN 2;
- 7: 5zqe-B MOLECULE: LMO2812 PROTEIN;
- 8: 6qvm-A MOLECULE: MULTIHEME CYTOCHROME CF;
- 9: 6rgv-A MOLECULE: FLAGELLIN;
- 10: 5zqe-G MOLECULE: LMO2812 PROTEIN;
- 11: 1tvf-A MOLECULE: PENICILLIN BINDING PROTEIN 4;
- 12: 3hun-A MOLECULE: PENICILLIN-BINDING PROTEIN 4;
- 13: 5tw8-A MOLECULE: PENICILLIN-BINDING PROTEIN 4;
- 14: 6dz8-B MOLECULE: PENICILLIN-BINDING PROTEIN 4;
- 15: 2nda-A MOLECULE: MID-CELL-ANCHORED PROTEIN Z;
- 16: 5txi-B MOLECULE: PENICILLIN BINDING PROTEIN 4;

- 17: 5zqe-D MOLECULE: LMO2812 PROTEIN;
18: 1xp4-D MOLECULE: D-ALANYL-D-ALANINE CARBOXYPEPTIDASE;
19: 1mgw-A MOLECULE: GUANYL-SPECIFIC RIBONUCLEASE SA3;
20: 6rf1-I MOLECULE: DNA-DEPENDENT RNA POLYMERASE SUBUNIT RPO132;

We have now included a summary of the DALI search result in lines 142-146 of the manuscript.

(12) The Hermes octamer forms a ruler that defines (roughly) the LE and RE sequences that are captured by the non-catalytic C dimer. How does this sequence compare to the TIR sequence and how is Hermes prevented from cleaving them?

As the reviewer correctly comments, the interaction of the transposon sequences with an unused catalytic domain could lead to a nick or even a double-strand break that would be clearly destructive for the transposon. Below, we show the alignment of the LE-TIR (top) with the 3'-end sequences of the DNAs used for the cryo-EM (the base pairs in gray are not present in the oligo used in the experiments but they are the true transposon end sequences). The sequences show very little similarity to the TIR, and this might be the reason why they would not be nicked or cleaved. We have included a statement about this issue in lines 293-296 of the manuscript.

```
LE-TIR: 5' -CAG1AGA5ACA10ACA15CAAG
3'-end LE-DNA: 5' -AGCAAGTGGCGCATAAG
3'-end RE-DNA: 5' -TGCTTATCTATGTGGCT
```

(13) The authors should explain earlier in the paper that parallel orientation of LE/RE is expected in the transpososome and explain why; don't wait for the discussion. Many general readers may miss this in Fig 4.

We thank the reviewer for pointing this out. We now state this immediately in the first paragraph of the section entitled "Three LE-STRs bind to BED domains within the transpososome" (lines 237-240).

(14) Line 259. Bridged by two dna duplexes (not oligos)

Thank you, this has been corrected.

(15) Line 336. Need a better section title. Readout is transposition, not formation of synaptic complex. Also, wording is difficult to follow.

The reviewer is correct that our original choice of words wasn't great. The section title has been changed to "*Restoring asymmetric BED binding sites in the LE/LE transposon does not rescue transposition activity in cells*" (lines 349-350).

(16) Line 411 - why is only binding to RE-TIR important? Doesn't binding by dimer C make a contribution as well?

The reviewer raises a good point. We have changed the sentence to indicate that TIR binding is "perhaps aided by non-specific contributions from dimer C across the octamer" (lines 442-443).

(17) Discussion - is a key role of nucleosomes likely in this case? The experimental readout is using pUC-like plasmids, which may not even have nucleosomes when Hermes synaptosomes form in HEK293 cells.

The reviewer is correct that there is a potential that nucleosomes may play a role in Hermes transposition. However, Gandaraham et al. looked at this issue (PNAS, 2010 Vol. 107, pp. 21966-21972) and observed that Hermes integrates into nucleosome-free regions in the genome. As subsequent mobilization would occur from these regions as well, we suggest that nucleosomes may not be a key player. Our recent experiments also shown that the STRs that are present further into the transposon DNA do not appear to play a critical role (Fig. 7E). It is also possible that some cellular factors - for instance ones that can bridge DNA - may play a role and we are of course very interested in pursuing these question. However, these efforts are beyond the scope of the current work.

REVIEWERS' COMMENTS

Reviewer #1 (Remarks to the Author):

The authors have attempted to improve the cryo-EM map by collecting more data with tilted compustage. Despite such efforts, the cryo-EM map was not improved significantly. Nevertheless, I think the overall modelling present in this work is still reliable, given that the crystal structures of the majority part of Hermes transposon have been determined. I therefore support the publication of this work at Nature Communications.

Reviewer #2 (Remarks to the Author):

Lannes and coworkers have fully addressed my comments and those from other reviewers. I particularly appreciate the additional analysis and clear explanation of the crystal structure, as well as the thorough examination of the cryo-EM data. This work is a seminal contribution to our understanding of Hermes

Reviewer #3 (Remarks to the Author):

The authors have done an excellent job in responding to reviewer comments in my opinion. The paper should be published.